# ACTION-FREE OFFLINE-TO-ONLINE RL VIA DISCRETISED STATE POLICIES

**Natinael Solomon Neggatu**\*, **Jeremie Houssineau**†, **& Giovanni Montana**\*‡
{nati.neggatu, g.montana}@warwick.ac.uk,
jeremie.houssineau@ntu.edu.sg

## ABSTRACT

Most existing offline RL methods presume the availability of action labels within the dataset, but in many practical scenarios, actions may be missing due to privacy, storage, or sensor limitations. We formalise the setting of action-free offline-to-online RL, where agents must learn from datasets consisting solely of $(s, r, s')$ tuples and later leverage this knowledge during online interaction. To address this challenge, we propose learning state policies that recommend desirable next-state transitions rather than actions. Our contributions are twofold. First, we introduce a simple yet novel state discretisation transformation and propose Offline State-Only DecQN (OSO-DecQN), a value-based algorithm designed to pre-train state policies from action-free data. OSO-DecQN integrates the transformation to scale efficiently to high-dimensional problems while avoiding instability and overfitting associated with continuous state prediction. Second, we propose a novel mechanism for guided online learning that leverages these pre-trained state policies to accelerate the learning of online agents. Together, these components establish a scalable and practical framework for leveraging action-free datasets to accelerate online RL. Empirical results across diverse benchmarks demonstrate that our approach improves convergence speed and asymptotic performance, while analyses reveal that discretisation and regularisation are critical to its effectiveness.

## 1 INTRODUCTION

Offline reinforcement learning (RL) focuses on learning effective policies entirely from a fixed dataset gathered by an unknown behaviour policy, avoiding the need for risky or costly online exploration. While most offline RL research assumes that all necessary training information is readily available in the dataset, in this work, we explore the scenario where action data is completely absent. Such missing or corrupted actions can occur due to privacy requirements, sensor failures, or simply because actions were never logged. For example, in healthcare settings (Kushida et al., 2012; Yu et al., 2021), explicit treatment decisions may be excluded from patient records for legal or confidentiality reasons, leaving only patient states (symptoms, vitals, lab results). In finance (Liu et al., 2022; Loukides & Gkoulalas-Divanis, 2012), the exact trades or order flows might be withheld to protect proprietary strategies, so only partial portfolio states are available. In robotics or industrial control, sensor logs are sometimes kept without recording the precise torque or command signals, either to reduce storage overhead or because that information is considered proprietary. In these scenarios, agents must learn from action-free datasets that contain only $(s, r, s')$ tuples. This raises a fundamental question: *can we learn effective policies and transfer them to online RL without ever observing offline actions?* We call this setting action-free offline-to-online RL and propose a scalable framework to address it by learning state policies over next-state transitions rather than actions.

Recent methods address parts of this problem but leave fundamental limitations: Earlier Q-value formulations over target states (Edwards et al., 2020; Hepburn et al., 2024) require actions during training, preventing their application in action-free settings. Transformer-based state policies (Zhu et al., 2023) attempt to remove this dependency, but they incur high computational costs and have

---

\*University of Warwick
†Nanyang Technological University
‡The Alan Turing Institute

not demonstrated consistent improvements when guiding online RL across diverse environments and dataset qualities. Moreover, existing discretisation strategies (Seyde et al., 2022) have been designed for actions with bounded ranges, making them unsuitable for action-free datasets. None of these works has demonstrated a practical scalable approach that can (a) learn from action-free offline data, (b) scale to high-dimensional control problems and (c) be used for guided online learning to accelerate online performance across a wide range of environments.

In this work, we structure our contribution into two complementary stages. First, we present a simple yet effective strategy for pre-training offline state policies; then we propose a pipeline that leverages these policies for guided online learning that addresses all three criteria.

For our pre-training strategy, we introduce a novel state discretisation transformation and offline RL algorithm, Offline State-Only DecQN (OSO-DecQN), designed to learn state policies that predict discretised state differences from action-free datasets. The algorithm integrates our discretisation mechanism with conservative regularisation adapted to the discrete state domain. This design avoids the instability of continuous regression, mitigates overestimation bias, and constrains predictions to reachable states. Unlike Seyde et al. (2022), our discretisation method is inherently scale-invariant, facilitating learning across state dimensions with varying numerical ranges.

This leads to our second contribution, where we develop a novel mechanism that leverages the pre-trained state policy to accelerate online RL. This mechanism introduces:

- An **online-trained inverse dynamics model (IDM)** to translate predicted state differences ($\Delta s$) into executable actions, trained from scratch and learned jointly with the online agent.
- A **policy-switching strategy** that blends offline-guided actions with the online agent's own policy, enabling smooth integration of offline knowledge without destabilising learning.

Through extensive experiments across a diverse range of tasks and datasets of varying quality, we show that our approach can effectively guide exploration online, improving the performance of online algorithms beyond existing action-free methods on a wide range of tasks, including environments with up to 78 state dimensions. To our knowledge, no prior work has combined an action-free offline state policy learner with a mechanism to guide online learning to produce consistent gains across diverse control environments.

To better understand why our approach provides valuable signals for enhancing online learning, we undergo a novel offline analysis comparing our method to alternative algorithms that pre-train state policies. Through this comparative analysis, we reveal that:

- The naive approach of training state policies using continuous next-state targets results in state policies with poor prediction capabilities.
- Our proposed discretisation mechanism enables us to sidestep the difficulties inherent in continuous regression.
- Despite the loss of information via discretisation, OSO-DecQN is capable of suggesting trajectories that outperform the empirical performance of the dataset.
- Regularisation is a critical component in OSO-DecQN to prevent overestimation bias and ensure prediction error of state policies is small.

## 2 RELATED WORK

**Offline RL**    Offline RL methods focus on learning policies using a fixed dataset generated by an unknown behaviour policy. A large proportion of methods focus on constraining policies to stay close to the support of the data during training with the aim of mitigating the issues of distributional shift. Although this has been achieved in various ways in the literature, most methods fall into two broad categories. Policy-constraint methods (Fujimoto & Gu, 2021; Wu et al., 2019; Fujimoto et al., 2019; Kumar et al., 2019) encompass the set of algorithms that look to explicitly constrain the policy to ensure that the agent stays close to the data. Alternatively, conservative methods (An et al., 2021; Bai et al., 2022; Kumar et al., 2020) look at penalising the Q-value estimate of out-of-distribution (OOD) actions. OSO-DecQN extends these concepts to the action-free setting, adapting conservative

regularisation techniques to constrain state transitions rather than actions, that additionally addresses the challenge of state reachability that is unique to action-free offline RL.

**State-only RL** Edwards et al. (2020) develop a novel form of value function, QSS', which estimates the return of transitioning to a target state rather than a selected action. Despite estimating the Q-value of transitioning to next-states, they rely on actions to make next state predictions that can be used to train the critic. Hepburn et al. (2024) extend this method to the offline setting, positing that it can be more effective to develop policies from offline data by estimating the critic values of target states instead of actions. However, like Edwards et al. (2020), they still rely on using actions to train their networks. In contrast, Zhu et al. (2023) removes the need for action labels entirely and instead uses decision transformers to learn a state policy from action-free datasets offline. They build off the work of Chen et al. (2021) and use entire trajectories instead of a single state to estimate the next state in the sequence. Decision transformers have also been proposed in an offline setting when actions are available (Bhargava et al., 2024), but several offline algorithms (Fujimoto & Gu, 2021; An et al., 2021; Beeson & Montana, 2024) outperform them while also being less computationally expensive. Other action-free approaches adapt transformers (Zhou et al., 2023; 2024; Luo et al., 2024) for visual data and unsupervised pretraining (Seo et al., 2022). However, Zhou et al. (2023); Luo et al. (2023); Seo et al. (2022), assume that the visual demonstrations are of expert or near-expert quality, which enables purely imitation-learning objectives. In contrast, in our offline RL setting, the data generating process is of unknown (and potentially mixed) quality. Zhu et al. (2023) can be viewed as an adaptation of Zhou et al. (2023) to the offline RL setting, using a similar implicit reward mechanism to guide online agents. Furthermore, our work differs by employing a simple discretisation strategy that circumvents the need for computationally intensive transformer-based architectures, while providing more stable performance through effective regularisation that addresses the critical reachability constraints overlooked in previous work.

**Single agent action value decomposition** Seyde et al. (2022) investigate solving continuous state-action problems using algorithms designed for discrete action settings. They develop a novel algorithm, Decoupled Q-Networks (DecQN), that combines deep Q-learning (Mnih et al., 2015) with value decomposition (Sunehag et al., 2018; Rashid et al., 2020), studied in multi-agent RL (MARL), to learn utility values for each action dimension. In their work, they treat each action dimension independently, allowing them to reduce the computational complexity from scaling exponentially with the action dimension to linearly. Ireland & Montana (2024) build on DecQN by showing that the use of ensembles can reduce variance, improving sample efficiency. Beeson et al. (2024) explore the use of DecQN in the offline setting for continuous action environments. They generate several discrete action datasets across a wide range of continuous-action tasks to benchmark offline variants of DecQN. Tang et al. (2022) also look at using a linearly decomposed critic for offline RL, although they focus solely on discrete action environments. These efforts collectively show that decomposing the critic can be highly effective in both online and offline scenarios, which motivates our extension of decomposed Q-learning to action-free datasets.

## 3 BACKGROUND

### 3.1 PRELIMINARIES

We model the RL problem as a Markov decision process (MDP) that can be completely specified by the tuple $(\mathcal{S}, \mathcal{A}, P, r, \rho, \gamma)$, where $\mathcal{S} \subseteq \mathbb{R}^M$ and $\mathcal{A} = \prod_{i=1}^N \mathcal{A}_i \subseteq \mathbb{R}^N$ are, respectively, the state and action space. We assume the continuous action space has this Cartesian product structure, with no additional coupling constraints between dimensions. $P(s'|s, a)$ defines the environment dynamics, $\rho(s_0)$ describes the initial state distribution, $r : \mathcal{S} \times \mathcal{A} \to \mathbb{R}$ is the reward function, and $\gamma \in (0, 1]$ is the discount factor. The agent's behaviour within the environment is specified by policy $\pi(a|s)$ and the expected return for a given policy can be described using the action value function $Q^\pi(s, a) = \mathbb{E}_\pi[\sum_{t \geq 0} \gamma^t r_t | s_0 = s, a_0 = a]$.

### 3.2 DECOUPLED Q-NETWORKS

Both Seyde et al. (2022) and Tang et al. (2022) propose the idea of decomposing the Q function, used in Deep Q-Networks (DQN) (Mnih et al., 2015), into utility functions $U^i(s, a_i)$ for each action

dimension. The overall action value is then estimated as the average over the utility functions as

$$Q_\theta(s, \mathbf{a}) = \frac{1}{N} \sum_{j=1}^N U_{\theta_j}^j(s, a_j) \tag{1}$$

where $\mathbf{a} = (a_1, \dots, a_N) \in \mathbb{R}^N$ and $U^j$ computes the utility value for the $j^{th}$ action dimension. This approach effectively treats each action dimension independently, reducing what would otherwise be an exponential complexity in discrete action settings to linear in $N$.

We define the argument of the maximum over $\mathbf{a}$ in an element-wise fashion as

$$\arg\max_{\mathbf{a}} Q_\theta(s, \mathbf{a}) := \left( \arg\max_{a_1} U_{\theta_1}^1(s, a_1), \dots, \arg\max_{a_N} U_{\theta_N}^N(s, a_N) \right)$$

The Q function is then trained by minimising the following loss function

$$\mathcal{L}(\theta) = \frac{1}{|B|} \sum_{(s_0, \mathbf{a}_0, r_{0:n-1}, s_n) \in B} L(y^n - Q_\theta(s_0, \mathbf{a}_0)) \tag{2}$$

where $B$ is the sampled batch and $y^n = \sum_{j=0}^{n-1} \gamma^j r_j + \gamma^n Q_{\bar\theta}(s_n, \mathbf{a}_n^\star)$ is the $n$-step Q-learning target, with $\bar\theta$ the lagged parameter of the target Q network. The action $\mathbf{a}^\star$ can be retrieved efficiently as $\mathbf{a}^\star = \arg\max_{\mathbf{a}} Q_{\bar\theta}(s, \mathbf{a})$. The most common choices of $L$ have been the Huber and MSE functions.

## 4 METHODOLOGY

### 4.1 DISCRETISING THE TARGET STATE SPACE

Our goal for pre-training is to learn a state policy that predicts how each state dimension should evolve to yield high returns. Instead of directly regressing to continuous next states, an approach prone to instability and overfitting, we introduce a simple discretisation transformation. For each state dimension, we learn whether it will increase, decrease, or remain unchanged. Formally, we define $\delta^\epsilon(s, s')$ as a vector in $\{-1, 0, 1\}^M$ with $i$-th component

$$\delta_i^\epsilon(s, s') = \begin{cases} -1 & \text{if } s_i' - s_i < -\epsilon \\ 1 & \text{if } s_i' - s_i > \epsilon \\ 0 & \text{otherwise.} \end{cases} \tag{3}$$

This transformation enables us to replace the instability of continuous regression with a discrete prediction problem. In turn, it allows us to leverage the well-studied value-based Q-learning framework to effectively learn from action-free datasets. We henceforth use the shorthand notation $\Delta s$ to denote the discretised state difference.

Unlike the action discretisation approach of Seyde et al. (2022), we discretise state differences after first applying z-score normalisation to each state dimension, yielding an effectively scale-invariant transformation. This eliminates dependence on raw numerical ranges and avoids costly per-dimension tuning, leading to a more robust performance across heterogeneous state spaces.

Discretising state differences simplifies prediction but inevitably introduces approximation error. To quantify this effect, we establish the following bound.

**Theorem 1** (Value bound under $k$ evenly spaced bins in $\Delta s$). *If the discretised increment space $\Delta s$ is partitioned into $k$ evenly spaced bins per coordinate, then*

$$\left\| V^* - V_D^* \right\|_\infty = O\left( \frac{H\sqrt{M}}{k} \right).$$

*where $V^*$ and $V_D^*$ are the optimal value functions of the original and discretised MDPs. Here $M$ denotes the state dimension and $H := \delta_s^{\max} - \delta_s^{\min}$ denotes the range of the per-coordinate mean increments. The required assumptions, and the proof are provided in Appendix B.*

This bound shows that discretising each state dimension into a finite number of bins preserves the value function up to a controllable error that diminishes as the discretisation becomes finer.

## 4.2 OSO-DecQN

Our algorithm, Offline State Only DecQN (OSO-DecQN), adapts the DecQN framework to the action-free setting by replacing discrete actions with discretised state differences. Specifically, we define Q-values of the form $Q(s, \Delta s)$, where $\Delta s \in \{-1, 0, 1\}^M$ represents the discretised change in each state dimension. This reformulation allows us to estimate action-free Q-values directly from $(s, r, s')$ tuples, without requiring explicit action labels.

While DecQN decomposes Q-values across action dimensions, OSO-DecQN instead decomposes across state-difference dimensions, enabling the same scalability benefits in high-dimensional state spaces. We use an ensemble variant of DecQN for improved sample efficiency, and employ a variant of double Q-learning (Van Hasselt et al., 2016) to stabilise learning. Importantly, we further introduce a regularisation term $\mathcal{R}_\theta$, detailed in Section 4.3 that mitigates overestimation bias and addresses state reachability constraints, a challenge unique to pre-training from action-free learning and not addressed in standard DecQN.

---

**Algorithm 1** OSO-DecQN

---

1: Load dataset $\mathcal{D}$ with samples of the form $(s, r, s')$
2: **for** $i = 1 \ldots N$ **do**
3:     Sample mini batch of transitions $(s, r, s') \sim \mathcal{D}$
4:     Sample $\Delta s'$ based on the softmax of $Q_\theta(s', \cdot)$
5:     Compute target $y^1 = r + \gamma Q_{\bar\theta}(s', \Delta s')$
6:     Update parameters:
7:

$$\theta \leftarrow \arg\min_\theta \sum \left[ y^1 - Q_\theta(s, \Delta s) \right]^2 + \alpha \mathcal{R}_\theta$$

8: **end for**

---

## 4.3 REGULARISATION

Various regularisation techniques have been used in offline RL to constrain action policies from deviating away from the support of the data. Many such methods rely on the actor-critic framework that separates policy and critic learning. Value-based methods such as DecQN, however, implicitly learn the policy by training the critic, and thus cannot utilise such regularisation methods with the same convergence guarantees. Kumar et al. (2020) propose a penalty term in the critic loss to conservatively learn critic values in the actor-critic framework, which Luo et al. (2023) show is equivalent to the negative log-likelihood behavioural cloning (BC) loss in the discrete action setting. Denoting $\| \exp(Q_\theta(s, \cdot)) \|_1 = \sum_{\Delta s} \exp(Q_\theta(s, \Delta s))$, we adapt this regularisation term for our discrete state difference, $\mathcal{R}_\theta = \sum_{(s, \Delta s) \sim \mathcal{D}} \log \| \exp(Q_\theta(s, \cdot)) \|_1 - Q_\theta(s, \Delta s)$, in Algorithm 1, to mitigate the issues of both overestimation bias and state reachability constraints.

## 4.4 GUIDED ONLINE LEARNING

As offline trained state policies can not be directly used to interact with the environment, we instead explore how they can be used to guide online learning. In particular, we identify several key factors that must be taken into account for guided online learning. Namely, we consider how to leverage the offline Q-values that define the state policy to guide training online, which type of online agent to guide using our method, and finally how to train an inverse dynamics model (IDM), from scratch, concurrently with the online agent to translate next state predictions. We address each of these points in the following sections.

**Leveraging Q-values** To demonstrate how our offline algorithm can enhance online performance, we use its critic to guide exploration within a standard actor-critic algorithm. Specifically, we introduce a policy-switching mechanism that integrates the pre-trained state policy predictions into the online agent's decision-making, enabling more targeted and efficient exploration.

We use an IDM, denoted $I$, that is trained concurrently from scratch, to convert the state policy into actions and switch between the online policy and offline policy during training as follows:

$$\mathbf{a} = \begin{cases} I_\phi(s, \arg\max_{\Delta s} Q(s, \Delta s)) & \zeta < \beta \\ \pi_{\text{on}}(s) & \text{otherwise,} \end{cases}$$

where $\zeta \sim U(0, 1)$ and $\beta$ is a hyperparameter. The samples are stored in the online agent's replay buffer, where they are used to train both the IDM and the online agent.

Alternatively, Zhu et al. (2023) propose using an intrinsic guiding reward related to the discrepancy between the planned and true next state. However, without convergence guarantees, it is not clear whether this approach can generalise across tasks. Instead, we focus on leveraging Q-values to guide action selection and leave exploring reward shaping methods to future work.

**Training an IDM** We incorporate an IDM to translate next state predictions into actions that can be used to improve the online agent's behaviour. While more complex alternatives may exist for mapping next state predictions to actions, we deliberately adopt a minimal IDM as the simplest approach that integrates with our method. This choice ensures low computational overhead and keeps our focus on the state policy's contribution rather than the translator architecture.

To train the IDM, we use supervised learning. We find that the $L_1$ loss is better at achieving lower per-dimension errors than MSE. This is because the $L_1$ loss approximates the median whereas the $L_2$ loss approximates the mean, making the $L_1$ loss more robust to outliers. In addition to the samples obtained online, we also leverage samples from the offline dataset that we denote $(s_{\text{off}}, \Delta s_{\text{off}}, r_{\text{off}}, R_{\text{off}})$. Here, $R_{\text{off}}$ represents the observed discounted return. Since we assume that there are no actions in the offline dataset, we first assign an action using the online agent's policy as $\mathbf{a}_{\text{on,off}} = \pi_{\text{on}}(s_{\text{off}})$. We then determine what the IDM would predict by computing $I_\phi(s, \Delta s_{\text{off}})$. Consequently, the final loss is

$$L(\phi) = \|\mathbf{a}_{\text{on,off}} - I_\phi(s, \Delta s_{\text{off}})\|_1 + \|\mathbf{a} - I_\phi(s, \Delta s)\|_1.$$

**Mapping state discretisations to actions** Our guided online mechanism assumes that there exists a sufficiently regular inverse relationship between actions and local state transitions in the environment. In particular, for most states $s$ encountered online, the mapping from action to next state is locally smooth and predictable, so that a lightweight inverse dynamics model can learn $I_\phi(s, \Delta s)$ that approximately realises the target discretised transition $\Delta s$. This requires a local continuity in the dynamics (small changes in action induce coherent changes in the resulting state change) and a discretisation that is not so coarse as to collapse distinct next-state transitions induced by different actions into the same $\Delta s$. When these conditions fail, a more expressive IDM may be required to ensure that guidance benefits do not diminish. This can occur under strongly discontinuous dynamics or highly multi-modal inverse mappings where the same $(s, \Delta s)$ corresponds to many distinct actions and their smoothed average does not realise $\Delta s$.

## 5 EXPERIMENTAL RESULTS

In this section, we investigate the performance of OSO-DecQN at guiding online learning across a wide range of continuous and discrete action tasks. We pre-train the discrete state policies on the datasets from the D4RL suite (Fu et al., 2020) and the Action-factorinkedsed DeepMind Suite (Beeson et al., 2024), **removing all action information beforehand**. These tasks present a set of challenging control problems that allow us to assess the effectiveness and robustness of our algorithm. We provide more details on environments in Appendix C.1.

### 5.1 GUIDED ONLINE LEARNING

In Figure 1 we show that our method of guiding online agents using the offline trained state policies is capable of improving both asymptotic performance and convergence speed. This demonstrates the practical value of our action-free approach, as the learned state policies effectively transfer knowledge despite never having access to actions during offline training. We use TD3 (Fujimoto et al., 2018) as the agent we focus on improving for the continuous action datasets and an ensemble variant of DecQN that we denote DecQN_N for the discrete action datasets.

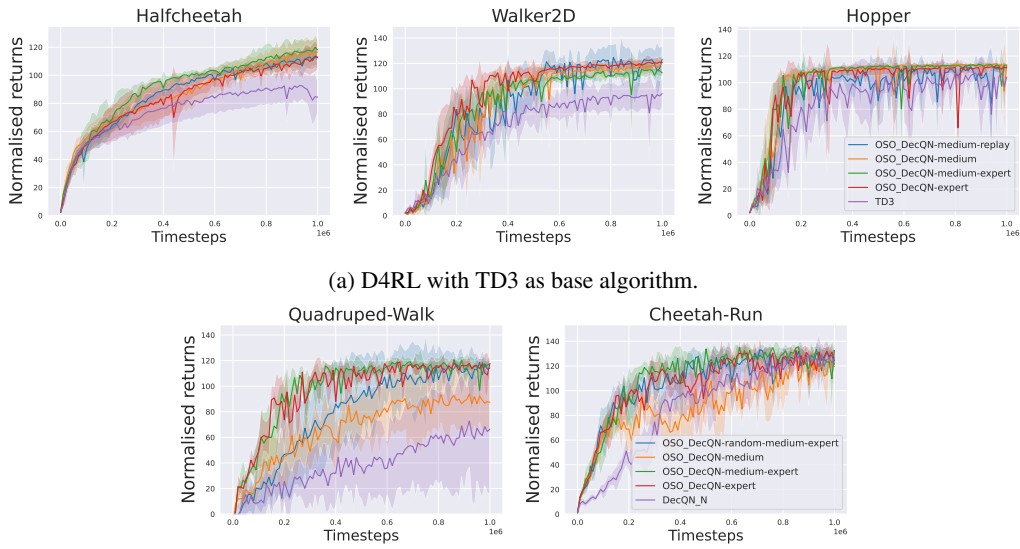

(a) D4RL with TD3 as base algorithm.

(b) DeepMind Control Suite with DecQN_N as base algorithm.

Figure 1: Online learning curves comparing the performance of guided online learning with state policies pre-trained on datasets of different qualities against baselines over 1M timesteps. The solid line corresponds to the mean normalised return across 5 seeds with the shaded area corresponding to 1 standard deviation away from the mean.

We observe that our method shows the smallest improvement in the hopper environment, which we believe is a result of being the simplest task with the fewest actions and observation dimensions, meaning the online agent is able to learn quickly without guided exploration. Despite this, we are still able to obtain slightly higher asymptotic performance and improve convergence speeds.

For Quadruped-Walk, the environment with the highest state dimension (78 in total), we achieve a consistent performance improvement during the early stages of learning, demonstrating the scalability of our approach. We observed that the performance of the online agent DecQN_N without guided learning was highly sensitive to the choice of seed, leading to variability in the results. As a result, the average performance is lower than the results reported in Ireland & Montana (2024). We found that this was the result of minor changes we made to the hyperparameters and architecture they provided, which led to increased variability in performance. Further details on our hyperparameter choices are provided in Appendix C. Although our main experiments focus on enhancing TD3, Appendix F demonstrates that the same method and set of hyperparameters can be applied to SAC with comparable improvements, showing that our approach can integrate seamlessly with other online algorithms and requires minimal tuning.

In conclusion, we observe that in both continuous and discrete action settings, OSO-DecQN is able to pre-train state policies that meaningfully accelerate and improve online learning, demonstrating that valuable knowledge can be extracted from offline datasets even without action labels.

**IDM analysis** To demonstrate the robustness of our method to the design of the IDM, we keep the IDM architecture and learning fixed across all environments in this section. Additionally, in Appendix E.1 we conduct an ablation study varying the IDM architecture and batch size. The results show that, across both smaller and larger architectures and batch sizes, performance remains similar. This indicates, in our tested settings, that the IDM has minimal effect on online guidance, allowing us to focus primarily on the predictive qualities of the state policy. In addition to these experiments, we also examine the sensitivity of guided learning to $\beta$ (Appendix D.3), demonstrating that our improvements persist across a range of hyperparameter values. Finally, we provide an empirical analysis of IDM learning dynamics (Appendix E.2). Together, these results suggest that, in our tested settings, the IDM can reliably translate state policy predictions across diverse environments while incurring minimal computational overhead.

**Benchmark comparison** We compare our method with that of Zhu et al. (2023) in Figure 2, which presents an alternative approach to guided online learning using state policies. To keep the plots uncluttered, we only include TD3 trained from scratch (we also provide a combined plot with all results in Figure 7) and note that, across all three environments, their method consistently underperforms relative to TD3, in contrast to ours. This further highlights the significance of our method in consistently providing a more effective signal for guiding online agents. Since our approach does not rely on a decision transformer, we can substantially reduce the time to pre-train state policies on action-free datasets. Furthermore, our algorithm scales efficiently with state dimensionality, enabling us to guide agents in environments with up to 78 state dimensions.

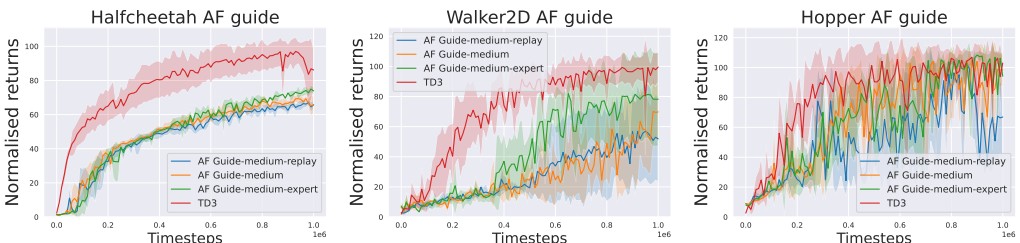

Figure 2: Comparison of Af-guide (Zhu et al., 2023) against TD3 baseline. The solid line corresponds to the mean normalised return across 5 seeds with the shaded area corresponding to 1 s.d.

# 6 PRE-TRAINING ANALYSIS AND ABLATION

In this section, we analyse our decision to discretise the state space and explain why the Q-values defining our state policy provide a valuable signal that improves online sample efficiency. We also examine the effect of regularisation on state-reachability constraints and overestimation bias.

**Inverse dynamics model (analysis only)** As we learn state policies from action-free datasets, we have no way to directly evaluate their performance in the environment. In this section, for the sole purpose of comparing the predictive capabilities of state policies pre-trained using different algorithms, we use an expert-level IDM to translate predicted next-state targets ($\Delta s$, $s'$, or $s' - s$) into actions so that they can be evaluated in the environment. By using an expert-level IDM, we minimise the bias introduced when mapping state policy predictions to actions, ensuring that observed differences in performance can be attributed to the state policies themselves. The expert IDM is trained independently of our method and all action-free baselines and is used **only** for the offline analysis in this section. It is not used anywhere in our main algorithm or online experiments.

**Why discretisation?** To understand why we discretise the state space, we investigate the performance of state policies trained using imitation learning. We use imitation learning to pre-train and compare the performance of state policies that predict the next state $s'$, the continuous state difference $s' - s$, and the discrete state difference that we denote $\Delta s$ across several datasets in the first four columns of Table 1 below. As a reference, we also provide the performance of a BC policy that is trained on the same dataset with actions. We denote the name of each imitation method as BC + the target it is trying to mimic.

Table 1 shows that the policy trained to mimic $\Delta s$ targets is capable of generating trajectories that yield returns that match the performance of an imitation policy trained with action labels. In contrast, directly mimicking the continuous target state or state difference from the dataset results in poor evaluation performance.

To further understand why performance is poor when directly imitating $s'$ or $s' - s$, we introduce an additional evaluation metric: the **discrete state difference error per step**, averaged over an entire trajectory. This metric, reported in Table 2 (Appendix A), complements the return-based evaluation in Table 1 by directly measuring the predictive accuracy of state policies. To compute this metric, we transform the true state difference $s'_{obs} - s_{obs}$, into a discrete state difference using the procedure outlined in Equation 3 and compare it to the predictions, $\Delta s_{predict}$, of the imitation policies at each time step during an episode. We compute the discrete difference error, $|\Delta s_{obs} - \Delta s_{predict}|$, for each

time step and average the error across an entire episode for each imitation learning algorithm. We expect that for algorithms that fail to replicate trajectories, this error will be much higher. For a random policy, the average discrete difference error is roughly equal to the number of dimensions in the state space, since the average error per state dimension is approximately 1. We include the discrete difference error for a random policy in Table 2 as a reference.

Table 1: Normalised average returns on D4RL and Factorised action tasks. Scores are averaged across 5 seeds with 10 episodes per seed. Ensemble-based methods use $N = 5$. Where relevant, we report the mean $\pm$ standard error.

| Dataset | Reference | Why discretisation? | | | Why regularisation? | |
|---|---|---|---|---|---|---|
| | BC $a$ | BC $s'$ | BC $s' - s$ | BC $\Delta s$ | DecQN_N | OSO-DecQN |
| Hopper | | | | | | |
| -medium-replay | 26.6 | 5.8 | 4.9 | $29.2 \pm 3.7$ | $7.7 \pm 1.3$ | $\mathbf{65.7 \pm 2.6}$ |
| -medium | 51.5 | 3.5 | 31.4 | $47.5 \pm 1.2$ | $1.2 \pm 0.31$ | $\mathbf{55.8 \pm 1.3}$ |
| -medium-expert | 51.3 | 2.9 | 15.6 | $51.9 \pm 2.1$ | $1.1 \pm 0.33$ | $\mathbf{110 \pm 1.3}$ |
| -expert | 110.6 | 2.2 | 9.9 | $106.9 \pm 2.4$ | $1.9 \pm 0.45$ | $\mathbf{111.6 \pm 0.08}$ |
| Halfcheetah | | | | | | |
| -medium-replay | 36.3 | -0.9 | -0.58 | $40.2 \pm 0.43$ | $-0.72 \pm 0.2$ | $\mathbf{43 \pm 0.54}$ |
| -medium | 41.9 | -0.2 | -0.41 | $42.8 \pm 0.21$ | $-1.9 \pm 0.15$ | $\mathbf{44.6 \pm 0.11}$ |
| -medium-expert | 60.1 | -0.25 | -0.25 | $54.7 \pm 3.9$ | $-1.6 \pm 0.13$ | $\mathbf{87.8 \pm 2.7}$ |
| -expert | 92.6 | -1.7 | -1 | $\mathbf{95.4 \pm 0.63}$ | $-1.9 \pm 0.09$ | $\mathbf{95.1 \pm 0.26}$ |
| Walker2d | | | | | | |
| -medium-replay | 23.5 | 6.4 | -0.32 | $37.5 \pm 6.1$ | $-0.41 \pm 0.26$ | $\mathbf{84.8 \pm 2.2}$ |
| -medium | 71.8 | 9.1 | -0.76 | $60.1 \pm 4$ | $-0.33 \pm 0.01$ | $\mathbf{77 \pm 1.3}$ |
| -medium-expert | 107.7 | 2.3 | -0.71 | $84.4 \pm 3.4$ | $-0.24 \pm 0.06$ | $\mathbf{108.8 \pm 0.13}$ |
| -expert | $\mathbf{108.2}$ | 17.1 | -0.69 | $108 \pm 0.12$ | $0.86 \pm 0.94$ | $\mathbf{108.1 \pm 0.13}$ |
| Cheetah-Run | | | | | | |
| -random-medium-expert | $\mathbf{41.5}$ | -0.16 | -0.16 | $36 \pm 1.6$ | $1.2 \pm 0.09$ | $40.2 \pm 0.8$ |
| -medium | 40.4 | -0.69 | -0.69 | $41.2 \pm 0.4$ | $0.95 \pm 0.13$ | $\mathbf{41.8 \pm 0.49}$ |
| -medium-expert | 61.6 | 1.7 | 1.7 | $48 \pm 3.4$ | $1.2 \pm 0.32$ | $\mathbf{90 \pm 3.8}$ |
| -expert | $\mathbf{99.9}$ | -0.56 | -0.56 | $93.6 \pm 3.7$ | $2.2 \pm 0.02$ | $\mathbf{98.4 \pm 1.9}$ |
| Quadruped-Walk | | | | | | |
| -random-medium-expert | 28 | 6.6 | 6.6 | $39.6 \pm 6.3$ | $2.3 \pm 0.1$ | $\mathbf{45.4 \pm 6.8}$ |
| -medium | 39.2 | 8.4 | 8.4 | $47.8 \pm 7.2$ | $2.9 \pm 0.32$ | $\mathbf{52.7 \pm 7.8}$ |
| -medium-expert | 63.4 | 4.7 | 4.7 | $84.7 \pm 8.1$ | $0.19 \pm 0.03$ | $\mathbf{91.2 \pm 5.8}$ |
| -expert | 97.7 | 6.6 | 6.6 | $96.7 \pm 6.4$ | $0.1 \pm 0.04$ | $\mathbf{100.7 \pm 1.2}$ |

Comparing the two tables, we see a strong correlation between the performance in Table 1 and predictive accuracy in Table 2 of state policies. Specifically, the imitation policies trained using $s'$ and $s' - s$ have errors comparable to the random policy on several benchmarks, whilst our transformation procedure allows us to keep errors down across all datasets. We also observe that the errors produced by our method are marginally larger for the more diverse datasets, medium-replay and medium-expert. This is a result of the imitation policy averaging across a wider range of discrete next states available in these datasets.

**Why regularisation?** We conduct an ablation study to understand the impact of regularisation. To do this, we compare our algorithm, OSO-DecQN, with a state-only variant of DecQN that is identical in all respects except that it omits the regularisation term. To control for ensemble size, we configure the benchmark to use the same number of critics as our method and refer to this variant as DecQN_N. We also compare against the imitation learning methods discussed in the previous section to demonstrate that our method is capable of using reinforcement learning to improve upon supervised learning methods.

In the last two columns of Table 1, we clearly see that OSO-DecQN is capable of outperforming both action and discrete state imitation learning methods. We believe this is a clear sign that the loss of information from our state discretisation transformation has minimal impact on our method's ability to use reinforcement learning to improve on the dataset trajectories. We also see that DecQN_N struggles to learn, indicating that regularisation is critical for offline performance and mitigating overestimation bias. In Table 2, we observe that the lack of regularisation in DecQN_N also results

in the trained state policy not being able to accurately predict next states, indicating the policy also suffers from state reachability constraints.

**Summary of findings**    From our analysis, we can conclude that both discretisation and regularisation are key components to ensuring that we can pre-train state policies effectively from action-free datasets. Table 2 shows that OSO-DecQN can accurately predict the next state, addressing state reachability constraints, whilst Table 1 shows that OSO-DecQN is capable of suggesting next states that yield high long-term returns, addressing overestimation bias. This explains why state policies trained using our method provide beneficial signals for guided online learning that result in enhanced sample efficiency.

In addition to the above analysis, Appendix D presents an ablation study on the sensitivity of pre-training performance to the discretisation threshold $\epsilon$ and a comparison between 2-bin and 3-bin discretisation. These results demonstrate that our method is robust to a broad range of $\epsilon$ values and discretisation granularities.

## 7    DISCUSSION

Leveraging datasets without action labels presents a fundamentally challenging problem where standard methods cannot be directly applied, yet such scenarios are common in domains like healthcare, finance, and robotics. Our work establishes a scalable and principled framework for action-free offline-to-online RL, showing that effective policies can be trained directly from $(s, r, s')$ tuples and then successfully transferred to online agents. By discretising state differences with a scale-invariant transformation and applying conservative regularisation, OSO-DecQN avoids the instability of continuous regression and constrains predictions to reachable states. Coupled with a lightweight guided online learning mechanism, this framework consistently improves convergence speed and asymptotic performance across diverse benchmarks, scaling to complex environments with up to 78 dimensions. These results provide the first evidence that reliable guidance can be extracted from action-free datasets at scale. Furthermore, our analysis in Section 6 demonstrates that our learned state policy provides useful signals for online guidance. We believe this analysis will be valuable to the community for benchmarking and validating alternative action-free state policies.

Looking forward, we see several promising directions for further developing our approach. One potential avenue could be to adopt adaptive discretisation mechanisms such as Seyde et al. (2024) to investigate whether more fine-tuned control can enhance pre-training performance and online guidance. Another is to evaluate more expressive translation mechanisms that go beyond the current minimal IDM, potentially improving robustness and performance in more complex domains.

Another promising avenue of research would be to adapt our method to learn from video comprised of suboptimal demonstrations similar to those available in vectorised state format in the D4RL and action-factorised datasets. While there exist methods that leverage existing encoders to transform images into vector representations compatible with our approach, additional work is needed on how to reliably extract reward information from visual data to execute RL algorithms.

In a complimentary direction, it would also be interesting to combine our method with recent work that replaces value-function regression by classification over discretised value supports (Farebrother et al., 2024). Incorporating such a cross-entropy based objective into our framework could further improve the stability and scalability of the critic, particularly when utilising larger architectures or solving problems in more complex domains.

ACKNOWLEDGEMENTS

This work was supported by the Engineering and Physical Sciences Research Council (2674942) and a UKRI Turing AI Acceleration Fellowship (EP/V024868/1). Jeremie Houssineau is supported by the Singapore Ministry of Digital Development and Information under the AI Visiting Professorship Programme, award number AIVP-2024-004.

## 8 REPRODUCIBILITY STATEMENT

We provide code as supplementary material to enable reproduction of all experiments presented in this paper. Detailed hyperparameter settings are provided in Appendix C. Additionally, we include all assumptions and complete proofs for the theorems stated in this work in Appendix B.

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

## A    STATE POLICY PREDICTION ERROR

Table 2: Average discrete difference error, $|\Delta s_{obs} - \Delta s_{predict}|$, across an entire episode. Results are averaged across 5 seeds with 10 episodes per seed. We report as reference the expected error of a random policy denoted "Rnd". Where relevant, we report the mean $\pm$ standard error.

| Dataset | Reference | Why discretisation? | | | Why regularisation? | |
|---|---|---|---|---|---|---|
| | Rnd | BC $s'$ | BC $s' - s$ | BC $\Delta s$ | DecQN_N | OSO-DecQN |
| Hopper | | | | | | |
| -medium-replay | 11 | 12.1 | 7.9 | $0.68 \pm 0.04$ | $11.8 \pm 0.54$ | $0.99 \pm 0.04$ |
| -medium | 11 | 10.5 | 1.9 | $0.25 \pm 0.01$ | $6.8 \pm 1.4$ | $0.18 \pm 0.008$ |
| -medium-expert | 11 | 10.8 | 6.9 | $0.18 \pm 0.02$ | $10.7 \pm 0.49$ | $0.16 \pm 0.004$ |
| -expert | 11 | 11.5 | 5.7 | $0.15 \pm 0.02$ | $7.2 \pm 1.3$ | $0.12 \pm 0.004$ |
| Halfcheetah | | | | | | |
| -medium-replay | 17 | 17.1 | 2.9 | $1.1 \pm 0.01$ | $11.1 \pm 0.2$ | $1.42 \pm 0.35$ |
| -medium | 17 | 15 | 3.5 | $0.39 \pm 0.004$ | $16.8 \pm 0.19$ | $0.76 \pm 0.01$ |
| -medium-expert | 17 | 17 | 2.3 | $0.46 \pm 0.04$ | $16.8 \pm 0.01$ | $0.43 \pm 0.02$ |
| -expert | 17 | 16.9 | 4.2 | $0.38 \pm 0.03$ | $16.9 \pm 0.36$ | $0.36 \pm 0.009$ |
| Walker2d | | | | | | |
| -medium-replay | 17 | 16.2 | 16.6 | $2.9 \pm 0.14$ | $17.3 \pm 0.81$ | $2.6 \pm 0.04$ |
| -medium | 17 | 15 | 16.6 | $1.8 \pm 0.3$ | $13.3 \pm 0.55$ | $1.6 \pm 0.02$ |
| -medium-expert | 17 | 20.3 | 16.6 | $0.87 \pm 0.08$ | $9.7 \pm 2.2$ | $0.75 \pm 0.02$ |
| -expert | 17 | 17.2 | 17.1 | $0.45 \pm 0.009$ | $14.4 \pm 0.48$ | $0.55 \pm 0.02$ |
| Cheetah-Run | | | | | | |
| -random-medium-expert | 17 | 16.8 | 6.5 | $2.6 \pm 0.06$ | $15.3 \pm 0.76$ | $1.89 \pm 0.01$ |
| -medium | 17 | 16 | 5.1 | $1.4 \pm 0.02$ | $16.7 \pm 0.13$ | $1.07 \pm 0.01$ |
| -medium-expert | 17 | 15.8 | 13.5 | $1.6 \pm 0.02$ | $12 \pm 0.63$ | $1.05 \pm 0.09$ |
| -expert | 17 | 16.9 | 4.1 | $1.2 \pm 0.09$ | $17 \pm 0.02$ | $0.86 \pm 0.06$ |
| Quadruped-Walk | | | | | | |
| -random-medium-expert | 78 | 74.4 | 19.8 | $18 \pm 0.39$ | $77.1 \pm 0.03$ | $15.8 \pm 0.17$ |
| -medium | 78 | 77.6 | 19.8 | $13 \pm 0.27$ | $77.2 \pm 0.21$ | $12 \pm 0.25$ |
| -medium-expert | 78 | 74.5 | 10.6 | $14.7 \pm 0.34$ | $78.2 \pm 0.26$ | $13.8 \pm 0.17$ |
| -expert | 78 | 77 | 7.8 | $13.5 \pm 0.49$ | $77.2 \pm 0.04$ | $12.2 \pm 0.06$ |

Table 2 compares the predictive capabilities of state policies trained using different algorithms.

## B    PROOFS

**Assumptions and Definitions.**    Throughout, we define and assume:

- $\mathcal{M} = (S, \mathcal{A}, P, r, \gamma)$ is an MDP with rewards bounded within $[r_{\min}, r_{\max}]$, and we define $\Delta_r := r_{\max} - r_{\min}$.
- The discount factor satisfies $\gamma \in [0, 1)$.
- For each action $a \in \mathcal{A}$, discretisation is defined by a projection $b : \mathcal{A} \to \mathcal{A}_D$ mapping a continuous action to its discrete bin. Without loss of generality, we assume this mapping is independent of the state.
- $\mathcal{M}_D = (S, \mathcal{A}_D, P, r, \gamma)$ denotes the MDP obtained from $\mathcal{M}$ by discretising its action space.
- $V^*$ and $V_D^*$ are the optimal value functions of $\mathcal{M}$ and its discretisation $\mathcal{M}_{\mathcal{D}}$ respectively.
- $\epsilon_{KL}$ is the symmetrised KL mismatch of next–state distributions and is defined as

$$\epsilon_{\mathrm{KL}} := \sup_{s \in S} \sup_{a \in \mathcal{A}} \min\Big\{D_{\mathrm{KL}}\big(P(\cdot|s,a) \,\|\, P(\cdot|s,b(a))\big),\ D_{\mathrm{KL}}\big(P(\cdot|s,b(a)) \,\|\, P(\cdot|s,a))\big)\Big\}.$$

**Lemma 1** (Control of expectation differences via TV and KL)**.**    *Let $\mu, \nu$ be probability measures on a measurable space and let $f$ be bounded and measurable. Then*

$$\left|\int f \, d\mu - \int f \, d\nu \right| \leq \mathrm{sp}(f)\,\mathrm{TV}(\mu,\nu), \qquad \mathrm{sp}(f) := \sup_x f(x) - \inf_x f(x),$$

*see Dudley (2002). Moreover, by Pinsker's inequality (Cover & Thomas, 2006),*

$$\mathrm{TV}(\mu, \nu) \;\leq\; \sqrt{\tfrac{1}{2}\, D_{\mathrm{KL}}(\mu \| \nu)}.$$

**Lemma 2** (KL bound for discretised actions). *If $\mathcal{M}$ is discretised by binning each action $a \mapsto b(a)$, then*

$$\|V^* - V_D^*\|_\infty \;\leq\; \frac{\Delta_r}{1-\gamma} \;+\; \frac{\gamma}{(1-\gamma)^2}\, \Delta_r \sqrt{\frac{\epsilon_{\mathrm{KL}}}{2}},$$

*Proof.* Let the Bellman operators be

$$(TV)(s) := \max_{a \in \mathcal{A}} \Big\{ r(s,a) + \gamma\, \mathbb{E}_{s' \sim P(\cdot|s,a)}[V(s')] \Big\},$$

and

$$(T_D V)(s) := \max_{\bar{a} \in \mathcal{A}_D} \Big\{ r(s,\bar{a}) + \gamma\, \mathbb{E}_{s' \sim P(\cdot|s,\bar{a})}[V(s')] \Big\}.$$

By $\gamma$-contraction of $T_D$,

$$(1-\gamma)\, \|V^* - V_D^*\|_\infty \leq \Big\| \max_{a \in \mathcal{A}} Q^*(\cdot, a) - \max_{\tilde{a} \in \mathcal{A}_D} Q^*(\cdot, \tilde{a}) \Big\|_\infty.$$

Fix $s$ and let $a^* \in \arg\max_a Q^*(s,a)$; set $\bar{a} = b(a^*)$ the bin representative. Then

$$\max_{a \in \mathcal{A}} Q^*(s,a) - \max_{\tilde{a} \in \mathcal{A}_D} Q^*(s, \tilde{a}) \leq Q^*(s, a^*) - Q^*(s, \bar{a}).$$

Expanding $Q^*$ gives

$$Q^*(s, a^*) - Q^*(s, \bar{a}) = r(s, a^*) - r(s, \bar{a}) + \gamma \Big( \mathbb{E}_{s' \sim P(\cdot|s,a^*)}[V^*(s')] - \mathbb{E}_{s' \sim P(\cdot|s,\bar{a})}[V^*(s')] \Big).$$

Define $\Delta_r$ such that $\Delta_r := r_{\max} - r_{\min}$. By the TV dual bound and Pinsker (Lemma 1):

$$\Big| \mathbb{E}_{s' \sim P(\cdot|s,a^*)}[V^*(s')] - \mathbb{E}_{s' \sim P(\cdot|s,\bar{a})}[V^*(s')] \Big| \leq \mathrm{sp}(V^*) \sqrt{\tfrac{1}{2} D_{\mathrm{KL}}(P(\cdot|s,a^*) \| P(\cdot|s,\bar{a}))}.$$

Taking suprema over $s$ and inserting the definition of $\epsilon_{\mathrm{KL}}$,

$$(1-\gamma)\|V^* - V_D^*\|_\infty \leq \Delta_r + \gamma\, \mathrm{sp}(V^*) \sqrt{\frac{\epsilon_{\mathrm{KL}}}{2}}.$$

Finally, apply $\mathrm{sp}(V^*) \leq \Delta_r/(1-\gamma)$ to conclude

$$\|V^* - V_D^*\|_\infty \leq \frac{\Delta_r}{1-\gamma} + \frac{\gamma}{(1-\gamma)^2}\, \Delta_r \sqrt{\frac{\epsilon_{\mathrm{KL}}}{2}},$$

as claimed. $\qquad\square$

**Lemma 3** ($\Delta s$–binning bound in continuous space). *If the continuous increment $\delta s$ is discretised into $\Delta s$ by a binning rule, then*

$$\|V^* - V_D^*\|_\infty \;\leq\; \frac{\Delta_r}{1-\gamma} \;+\; \frac{\gamma}{(1-\gamma)^2}\, \Delta_r \sqrt{\frac{\epsilon_{\mathrm{KL}}}{2}},$$

*Proof.* Let $\delta := s' - s \in \mathbb{R}^M$ and $\phi_s : \mathbb{R}^M \to \mathcal{B}$ be a binning rule. Define the discretised increment $\Delta s := \phi_s(\delta)$ and denote by $P(\Delta s \mid s, a)$ the induced distribution. For each state $s$, form bins $\mathcal{C}_s$ by the equivalence relation $a \sim_s a'$ iff $P(\Delta s \mid s, a) = P(\Delta s \mid s, a')$, and pick a representative $\bar{a}_C \in C$ for each $C \in \mathcal{C}_s$.

Construct the discretised MDP $\mathcal{M}_D$ by mapping each $a \in C$ to its representative $\bar{a}_C(s)$. Applying Lemma 2 with this mapping and the above definition of $\epsilon_{\mathrm{KL}}$ yields the claimed bound. $\qquad\square$

**Theorem 2** (Value bound under $k$ evenly spaced bins in $\Delta s$). *Assume the continuous increment satisfies*

$$\delta s \mid (s,a) \sim \mathcal{N}(\mu_\Delta(s,a), \Sigma_\Delta(s)), \qquad \Sigma_\Delta(s) \succ 0 \text{ independent of } a,$$

*and there exist constants $\delta_{\min} < \delta_{\max}$ such that, for every coordinate $i = 1, \ldots, M$ and all $(s,a)$,*

$$\delta_{\min} \;\leq\; \mu_\Delta^i(s,a) \;\leq\; \delta_{\max}.$$

*If the discretised increment space $\Delta s$ is partitioned into $k$ evenly spaced bins per coordinate, then*

$$\left\| V^* - V_D^* \right\|_\infty \;=\; O\!\left( \frac{H\sqrt{M}}{k} \right).$$

*Proof.* For each $s$, partition the mean–increment box

$$\left[ \delta s_{\min}, \delta s_{\max} \right]^M$$

into $k$ equal bins per coordinate (evenly spaced). For each bin $C$, choose the representative $\bar{a}_C$ so that $\mu_\Delta\!\left(s, \bar{a}_C\right)$ is the bin center. Let

$$H := \delta s_{\max} - \delta s_{\min}, \qquad r_{\text{cell}}^{\Delta s} := \frac{H}{2k}\sqrt{M}, \qquad \Lambda_\Delta := \sup_s \lambda_{\max}\!\left( \Sigma_\Delta(s)^{-1} \right).$$

with $\lambda_{\max}(\cdot)$ the largest eigenvalue of its argument.

By construction, for any $a \in C$,

$$\left\| \mu_\Delta(s,a) - \mu_\Delta\!\left(s, \bar{a}_C\right) \right\|_2 \;\leq\; r_{\text{cell}}^{\Delta s}.$$

Since the covariances are equal and independent of $a$,

$$D_{\text{KL}}\!\left( \mathcal{N}(\mu, \Sigma) \,\|\, \mathcal{N}(\mu', \Sigma) \right) = \tfrac{1}{2}(\mu - \mu')^\top \Sigma^{-1}(\mu - \mu') \;\leq\; \tfrac{1}{2}\lambda_{\max}(\Sigma^{-1})\, \|\mu - \mu'\|_2^2.$$

Apply this with $\mu = \mu_\Delta(s,a)$, $\mu' = \mu_\Delta(s, \bar{a}_C)$, $\Sigma = \Sigma_\Delta(s)$, and then take suprema over $a \in C$, $C$, and $s$ to obtain the symmetrised mismatch

$$\epsilon_{\text{KL}} \;\leq\; \tfrac{1}{2}\,\Lambda_\Delta \left( r_{\text{cell}}^{\Delta s} \right)^2 \;=\; \frac{\Lambda_\Delta}{8} \frac{MH^2}{k^2}.$$

**Plug into Lemma 3.** Theorem 1 gives

$$\left\| V^* - V_D^* \right\|_\infty \;\leq\; \frac{\Delta_r}{1-\gamma} \;+\; \frac{\gamma}{(1-\gamma)^2}\, \Delta_r \sqrt{\frac{\epsilon_{\text{KL}}}{2}} \;\leq\; \frac{\Delta_r}{1-\gamma} \;+\; \frac{\gamma}{4(1-\gamma)^2}\, \Delta_r \sqrt{\Lambda_\Delta}\, \frac{H\sqrt{M}}{k},$$

which grows at the claimed order. $\qquad\square$

## C    EXPERIMENTAL DETAILS

### C.1    ENVIRONMENT DETAILS

**D4RL suite**    The D4RL (Fu et al., 2020) suite comprises datasets generated across a set of tasks that use the MuJoCo (Todorov et al., 2012) physics engine. For each task, rewards are provided according to the agent's ability to learn precise control and balance of a robotic model whilst in motion. For each problem, several datasets are generated using different quality policies. The different quality data allow us to investigate how robust our algorithm is to the data-generating process.

**Action-factorised DeepMind suite**    Beeson et al. (2024) investigate offline learning in a continuous action environment with datasets generated using discrete actions. We use their suite as an opportunity to investigate how our method handles offline learning with smaller datasets and environments with larger state and action dimensions than those available in the D4RL suite.

**Dataset sizes**    Table 3 summarises the number of offline state-only samples used for each dataset. For datasets that share the same size across environments, we simply indicate the dataset quality (medium, medium-expert, expert).

| Dataset names | Dataset size |
|---|---|
| D4RL Hopper medium-replay | 402k samples |
| D4RL HalfCheetah medium-replay | 202k samples |
| D4RL Walker2D medium-replay | 302k samples |
| D4RL medium | 1M samples |
| D4RL medium-expert | 2M samples |
| D4RL expert | 1M samples |
| Action factorised random-medium-expert | 200k samples |
| Action factorised medium | 1M samples |
| Action factorised medium-expert | 2M samples |
| Action factorised expert | 1M samples |

Table 3: Number of offline state-only samples in each dataset used to pre-train the offline state policy.

Table 4: List of general hyperparameters used.

| | Hyperparameter | Value |
|---|---|---|
| General Hyperparameters | Optimiser | Adam |
| | Mini-batch size | 256 |
| | Target update rate | 1e-3 |
| | Gamma | 0.99 |
| | Hidden dim | 512 |
| | Hidden layers | 3 |
| | Activation function | ReLU |
| | discrete epsilon ($\epsilon$) | 1e-4 |
| Offline DecQN_N | Ensemble size | 5 |
| | Learning rate | 1e-4 |
| | n step returns | 1 |
| Online DecQN_N | Ensemble size | 5 |
| | Minimum exploration | 0.05 |
| | decay rate | 0.999 |
| | n step returns | 3 |
| | Learning rate | 5e-4 |
| Online TD3 | Ensemble size | 2 |
| | n step returns | 1 |
| | Policy noise | 0.2 |
| | Policy noise clipping | (-0.5,0.5) |
| | Policy update frequency | 2 |
| | Critic learning rate | 5e-4 |
| | Actor learning rate | 5e-4 |
| IDM | Learning rate | 1e-3 |
| | Mini-batch size | 512 |

## C.2 OSO-DECQN HYPERPARAMETERS

We provide a detailed list of hyperparameters used for our experiments in Tables 4 and 5.

## C.3 IMPLEMENTATION DETAILS

We run all tests on a single GeForce RTX 3080. We implement our algorithm with Python 3.10 using the pytorch framework. We run all offline experiments for 3 million time steps and all online experiments for 1 million time steps. In Tables 6 and 7 we provide the computational cost of running our method on the Walker2d/Cheetah environment for offline pre-training and online learning. In Table 8, we provide a comparison of total runtime cost for our method compared to baseline methods on all environments. We note slightly higher runtimes for our guided online method as a result of the overhead of concurrently train an IDM. For the online hyperparameter $\beta$, we start with the initial value of $\beta_{\mathrm{max}}$ and decrease the value every 100k timesteps by $\beta_{\mathrm{decrement}}$ until $\beta_{\mathrm{min}}$ is reached. We

Table 5: Regularisation ($\alpha$) weights used during offline learning.

| Dataset | $\alpha$ |
|---|---|
| **Hopper** | |
| -medium-replay | 3 |
| -medium | 8 |
| -medium-expert | 10 |
| -expert | 8 |
| **Halfcheetah** | |
| -medium-replay | 2 |
| -medium | 3 |
| -medium-expert | 8 |
| -expert | 8 |
| **Walker2d** | |
| -medium-replay | 10 |
| -medium | 8 |
| -medium-expert | 8 |
| -expert | 8 |
| **Cheetah-Run** | |
| -random-medium-expert | 5 |
| -medium | 5 |
| -medium-expert | 5 |
| -expert | 5 |
| **Quadruped-Walk** | |
| -random-medium-expert | 5 |
| -medium | 5 |
| -medium-expert | 5 |
| -expert | 5 |

fix $\beta_{\max} = 0.5$, $\beta_{\text{decrement}} = 0.1$ and $\beta_{\min} = 0$ across all environments and dataset qualities. In Table 5 we list the regularisation ($\alpha$) values used. In line with how the datasets were curated, for Q learning, we use an MSE loss for D4RL suite and Huber loss for the action-factorised DeepMind suite. The general hyperparameters in Table 4 apply to all network architectures unless specified otherwise. All networks use a standard feed-forward neural network architecture.

| | Runtime offline (s/epoch$^\star$) | Runtime offline (s/epoch$^\star$) | GPU Mem. (offline) (MiB) |
|---|---|---|---|
| OSO-DecQN (N=5) | | 44.9 | 710 |
| BC $\Delta s$ | | 28 | 542 |

$^\star$1 epoch = 10,000 gradient steps

Table 6: Computational cost of pre-training on Walker2d/Cheetah environments.

| | Runtime online (s/epoch$^\star$) | GPU Mem. online (MiB) |
|---|---|---|
| Guided learning (Section 4.4) | 88.1 | 638 |
| DecQN_N (N=5) | 52.7 | 578 |
| TD3 (N=2) | 51 | 540 |

$^\star$1 epoch = 10,000 gradient steps

Table 7: Computational cost for (guided) online learning on Walker2d/Cheetah environments.

## C.4 BASELINES

We used our own implementation for all algorithms used in the main paper besides Behavioural Cloning (BC) with actions. We report the results of BC with actions from Beeson et al. (2024) for the factorised action datasets.

|  | Hopper | HalfCheetah | Walker2D | Cheetah-Run | Quadruped-Walk |
|---|---|---|---|---|---|
| OSO-DecQN offline + online | 2.9 + 1.8 | 3.7 + 1.9 | 3.7 + 1.7 | 3.3 + 1.9 | 5 + 2.3 |
| DecQN_N (N=5) | - | - | - | 1 | 1.4 |
| TD3 (N=2) | 1.1 | 1.2 | 1.1 | - | - |

Table 8: Total cost for our method on all environments compared to baselines (in hours). For OSO-DecQN we separate time into (pretraining) + (online).

## D ABLATION STUDIES

### D.1 IMPACT OF CHANGING $\epsilon$

We provide analysis demonstrating the effect of varying the value of $\epsilon$ that specifies the range of values that are converted to $0$ in the discretisation process. We look at how performance varies when the threshold value is set to $\epsilon = (1 \times 10^{-3}, 1 \times 10^{-4}, 1 \times 10^{-5}, 1 \times 10^{-6})$

Table 9: Ablation studying the effects of varying $\epsilon$. Results averaged across 3 seeds $\pm$ 1 s.e.

|  | $1 \times 10^{-3}$ | $1 \times 10^{-4}$ | $1 \times 10^{-5}$ | $1 \times 10^{-6}$ |
|---|---|---|---|---|
| hopper medium expert | $107.5 \pm 7.3$ | $110 \pm 1.3$ | $112 \pm 0.09$ | $111.7 \pm 2.3$ |
| halfcheetah medium replay | $42.3 \pm 0.73$ | $43 \pm 0.54$ | $43.1 \pm 0.1$ | $41.2 \pm 1.25$ |
| halfcheetah medium expert | $93.2 \pm 0.77$ | $87.8 \pm 2.7$ | $93.8 \pm 2.3$ | $87.6 \pm 5.3$ |
| walker2d medium | $80.9 \pm 1.3$ | $77 \pm 1.3$ | $84.6 \pm 0.26$ | $79.7 \pm 2.8$ |
| walker2d expert | $109 \pm 0.24$ | $108.1 \pm 0.13$ | $108.8 \pm 0.12$ | $108.7 \pm 0.07$ |

### D.2 SETTING $\epsilon$ TO $0$

The parameter $\epsilon$ is introduced as a tolerance for declaring "no change" in a state dimension after normalisation. This helps avoid spurious sign flips caused by small numerical noise. In this section we look at the impact of removing 0 from the discrete options, instead discretising state difference into two bins instead of three. Despite only using two bins, the number of possible $\Delta s$ available from any given state is $2^{\dim(S)}$. We show in Table 10 that this is also a sufficient amount of granularity for our method to learn from an offline dataset. We do however observe some notable degradation in performance, namely Walker2d medium-replay and Hopper medium-replay indicating setting $\epsilon = 0$ slightly hurts some environments because every tiny fluctuation is treated as a change. We also observe, on average, higher variation across seeds. We do note minor improvements on performance in some datasets, most notably Walker2d-medium which we believe could be as a result of a better balance between overfitting and information loss for these less diverse datasets.

We stick to only investigating 2 or 3 bins, as these are the only obvious ways to discretise the state difference $s' - s$. Using 1 bin would result in no learning, and increasing the number of bins beyond 3 would require setting thresholds for each bin, which could likely have to vary for each state dimension.

Table 11 presents the average discrete difference error when using two and three bins. For reference, we include a column showing the expected error under a random policy for both bin configurations. We see that the error is lower when using fewer bins. Additionally, we report the discrete difference error of OSO-DecQN as a percentage of the corresponding random policy's error (in brackets), providing a normalised metric for comparison.

### D.3 EMPIRICAL ANALYSIS TO $\beta$ SENSITIVITY

In this section we analyse the sensitivity of our method to the choice of $\beta$ by comparing a linearly annealed schedule to keeping $\beta$ fixed throughout training

Table 10: Comparing the performance of our algorithm when discretising $\Delta s$ using two and three bins

| Dataset | BC $\Delta s$ (3 bins) | OSO-DecQN (2 bins) | OSO-DecQN (3 bins) |
|---|---|---|---|
| Hopper | | | |
| -medium-replay | $29.2 \pm 3.7$ | $61.5 \pm 7.27$ | $\mathbf{65.7 \pm 2.6}$ |
| -medium | $47.5 \pm 1.2$ | $52.2 \pm 1.31$ | $\mathbf{55.8 \pm 1.3}$ |
| -medium-expert | $51.9 \pm 2.1$ | $\mathbf{110.4 \pm 1.21}$ | $110 \pm 1.3$ |
| -expert | $106.9 \pm 2.4$ | $111.3 \pm 0.43$ | $\mathbf{111.6 \pm 0.08}$ |
| Halfcheetah | | | |
| -medium-replay | $40.2 \pm 0.43$ | $\mathbf{43.5 \pm 0.32}$ | $43 \pm 0.54$ |
| -medium | $42.8 \pm 0.21$ | $44 \pm 0.17$ | $\mathbf{44.6 \pm 0.11}$ |
| -medium-expert | $54.7 \pm 3.9$ | $86.3 \pm 1.57$ | $\mathbf{87.8 \pm 2.7}$ |
| -expert | $95.4 \pm 0.63$ | $\mathbf{97.9 \pm 0.26}$ | $95.1 \pm 0.26$ |
| Walker2d | | | |
| -medium-replay | $37.5 \pm 6.1$ | $73.8 \pm 3.8$ | $\mathbf{84.8 \pm 2.2}$ |
| -medium | $60.1 \pm 4$ | $\mathbf{81.7 \pm 0.76}$ | $77 \pm 1.3$ |
| -medium-expert | $84.4 \pm 3.4$ | $108.5 \pm 0.15$ | $\mathbf{108.8 \pm 0.13}$ |
| -expert | $108 \pm 0.12$ | $\mathbf{108.3 \pm 0.09}$ | $108.1 \pm 0.13$ |

Table 11: Average discrete difference error, $|\Delta s_{obs} - \Delta s_{predict}|$, across an entire episode. Results are averaged across 5 seeds with 10 episodes per seed. We report as reference the expected error of a random policy denoted "Rnd". In brackets we provide a percentage ratio of our algorithm error compared to the random policy using the respective number of bins. Where relevant, we report the mean $\pm$ standard error.

| Dataset | Rnd (3 bins) | Rnd (2bins) | OSO-DecQN (2 bins) | OSO-DecQN (3 bins) |
|---|---|---|---|---|
| Hopper | | | | |
| -medium-replay | 11 | 5.5 | $0.51 \pm 0.02$ (9.27%) | $0.99 \pm 0.04$ (9%) |
| -medium | 11 | 5.5 | $0.1 \pm 0.004$ (1.82%) | $0.18 \pm 0.008$ (1.64%) |
| -medium-expert | 11 | 5.5 | $0.07 \pm 0.004$ (1.27%) | $0.16 \pm 0.004$ (1.45%) |
| -expert | 11 | 5.5 | $0.06 \pm 0.001$ (1.09%) | $0.12 \pm 0.004$ (1.09%) |
| Halfcheetah | | | | |
| -medium-replay | 17 | 8.5 | $0.68 \pm 0.004$ (8 %) | $1.42 \pm 0.035$ (8.35%) |
| -medium | 17 | 8.5 | $0.35 \pm 0.002$ (4.12%) | $0.76 \pm 0.01$ (4.47%) |
| -medium-expert | 17 | 8.5 | $0.24 \pm 0.004$ (2.82%) | $0.43 \pm 0.02$ (2.53%) |
| -expert | 17 | 8.5 | $0.22 \pm 0.009$ (2.59%) | $0.36 \pm 0.009$ (2.12%) |
| Walker2d | | | | |
| -medium-replay | 17 | 8.5 | $1.48 \pm 0.02$ (17.4%) | $2.6 \pm 0.04$ (15.5%) |
| -medium | 17 | 8.5 | $0.81 \pm 0.02$ (9.52%) | $1.6 \pm 0.02$ (9.59%) |
| -medium-expert | 17 | 8.5 | $0.41 \pm 0.02$ (4.82%) | $0.75 \pm 0.02$ (4.41%) |
| -expert | 17 | 8.5 | $0.27 \pm 0.01$ (3.18%) | $0.55 \pm 0.02$ (3.24%) |

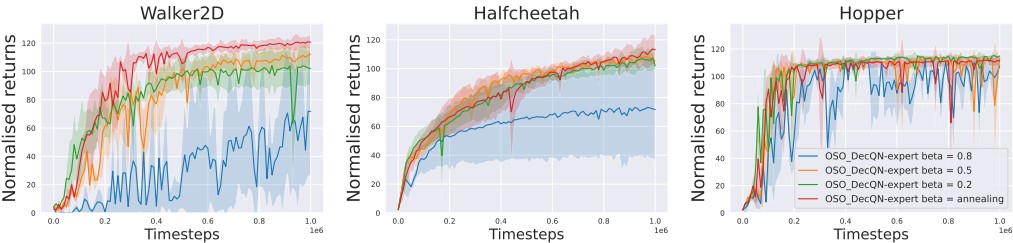

Figure 3: Comparison of fixed versus linearly annealed $\beta$ in guided online learning for Walker2D and Hopper.

In Figure 3, we observe that in the simpler Hopper environment, where the online agent converges rapidly, the choice of $\beta$ has the smallest impact on performance. In contrast, in both HalfCheetah and

Walker2D environments, performance degrades substantially when $\beta = 0.8$ is kept fixed throughout training. In these cases, a large proportion of actions originate from the offline state policy while the IDM is still inaccurate, leading to poor early exploration that hinders the online agent.

This degradation is mitigated when smaller $\beta$ values are used. For example, in HalfCheetah, performance with $\beta = 0.2$ and $\beta = 0.5$ is similar during learning, but annealing $\beta$ results in slightly improved asymptotic performance. In Walker2D, annealing from $\beta = 0.5$ yields the best results, combining strong early exploration from guided actions with the flexibility to rely more on the agent's own learned policy later in training.

Our interpretation is that $\beta$ controls the exploration-exploitation balance induced by the offline state policy. A larger $\beta$ increases the probability that the agent follows suggestions from the offline state policy rather than its current online policy, which encourages offline-guided exploration early in training but can lead to under-exploitation of the learned policy later on. However, when $\beta$ is too large, the offline guidance becomes dominated by noise in the IDM predictions, leading to the performance degradation observed in Walker2D and HalfCheetah.

In contrast, the annealed schedule starts from a relatively high $\beta$, leveraging the offline state policy to diversify exploration when the online agent is still weak, and then gradually reduces $\beta$ so that the agent increasingly exploits its own learned policy. This dynamic shift from offline-guided exploration to exploitation of the online policy improves both stability and final performance. While annealed $\beta$ initially learns slower than $\beta = 0.2$ (due to the noisy IDM predictions), it quickly surpasses $\beta = 0.2$ as the IDM becomes more accurate, achieving improved returns. This supports our view that annealing $\beta$ provides more favourable exploration-exploitation trade-off than any fixed choice.

## E  ONLINE IDM ANALYSIS

### E.1  ARCHITECTURAL AND BATCH SIZE ABLATION

In Table 12, we see that varying the IDM depth and minibatch size produced only a moderate change in guided online learning performance. While three layers with a batch size of 512 achieved the highest returns in both environments, the gains over other configurations were relatively small, indicating that the IDM capacity is not constraining learning performance. This supports our design choice of using a lightweight IDM, as performance is driven primarily by the quality of the offline state policy rather than fine-tuning performance.

Table 12: Effect of varying number of layers and minibatch size of the online IDM on final online performance for HalfCheetah medium and Walker2D medium. Results averages over 3 seeds $\pm$ 1 s.e.

| | HalfCheetah medium | | | | Walker2D medium | | |
|---|---|---|---|---|---|---|---|
| Layers | Batch 384 | Batch 512 | Batch 768 | Layers | Batch 384 | Batch 512 | Batch 768 |
| 2 | $117.9 \pm 5.9$ | $116.7 \pm 5.39$ | $113.6 \pm 11.8$ | 2 | $115 \pm 7.3$ | $113.3 \pm 3.2$ | $112.7 \pm 15$ |
| 3 | $111.6 \pm 5$ | $116.4 \pm 4.7$ | $116.4 \pm 8.2$ | 3 | $110.3 \pm 12.1$ | $120 \pm 5.1$ | $114.7 \pm 7.7$ |
| 4 | $103.2 \pm 2.1$ | $104.4 \pm 3.5$ | $114.4 \pm 6.3$ | 4 | $113.1 \pm 2.3$ | $116.8 \pm 6.9$ | $119.8 \pm 3.4$ |

Consistent with the observation, across IDM architectures we find only a weak correlation between IDM loss and final online return (Pearson $r = 0.19$ for HalfCheetah-medium and $r = -0.09$ for Walker2D-medium, indicating the variation in IDM prediction error explains little of the variation in performance.

### E.2  LEARNING DYNAMICS

In Figure 4 we look at the evolution of the online IDM loss, defined as:

$$L_{\text{IDM}} = \|\mathbf{a}_{\text{IDM,off}} - \mathbf{a}_{\text{on,off}}\|_1 + \|\mathbf{a} - I(s, \Delta s)\|_1$$

over the course of online training. Across all environments, the IDM loss converges quickly and remains stable for the remainder of learning. To quantify the final accuracy of the online IDM, Table 13 compares the final online IDM loss after training with the loss of the expert-level IDM used in

Section 6. We note that the online IDM loss is composed of two terms, making it roughly twice as large as the expert IDM loss. By considering approximately half of the total online IDM loss, the online IDM converges to a level comparable to the expert IDM across all environments. This demonstrates that, even without action-labelled pre-training, the online IDM can rapidly achieve expert-level mapping accuracy, ensuring it is not a limiting factor in guided online learning.

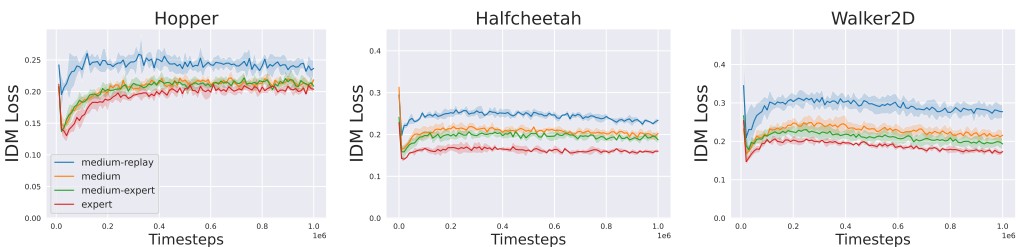

Figure 4: A look at the IDM loss function during online training

Table 13: Comparison of final online IDM loss and expert IDM loss. **The online IDM loss is the sum of two terms which makes it approximately twice as large as the expert IDM loss**. This shows that the online IDM loss converges to a similar performance as the expert IDM by the end of online learning. Results are averaged over 5 seeds $\pm$ 1 s.d.

| Dataset | Final expert IDM loss | Final online IDM loss |
|---|---|---|
| Hopper | 0.13 | $0.22 \pm 0.022$ |
| HalfCheetah | 0.12 | $0.196 \pm 0.028$ |
| Walker2D | 0.12 | $0.21 \pm 0.04$ |

We note that the IDM is included solely as a lightweight translator from $(s, \Delta s$ to actions for the purpose of online guidance, and is not part of OSO-DecQN itself. The loss curves here and $\beta$-sensitivity results (in Section 3) reported here demonstrate that our improvements are robust to variations in IDM learning speed and scheduling, reinforcing that the core performance gains stem from the offline state policy rather than the IDM's specific architecture or optimisation.

## F  GUIDANCE WITH SAC

In Figure 5 we demonstrate that our method enhances online learning when combined with SAC, compared to SAC without guidance, using the same hyperparameters as in Table 5. Since the hyperparameters were tuned primarily for TD3, our goal was to test whether the method can still guide SAC without additional tuning, underscoring its robustness.

Unlike TD3, which rapidly learns near-optimal behaviour in the Hopper environment even without guidance, SAC learns more slowly, making the benefits of guidance in accelerating learning more pronounced. Consistent with the results for TD3, we also observe that guidance improves both convergence speed and asymptotic performance in HalfCheetah and Walker2D when applied with SAC.

## G  COMPARISON TO OFFLINE-TO-ONLINE METHODS WITH ACTION LABELS.

In Figure 6 we compare OSO-DecQN to strong offline-to-online baselines that use action labels (O3F (Mark et al., 2022), IQL(Kostrikov et al., 2021), CAL-QL (Nakamoto et al., 2023), AWAC (Luo et al., 2023)) on Walker2D. As a result of training an action policy offline offline-to-online methods start online fine-tuning with a much higher initial performance than guidance with state-only datasets.

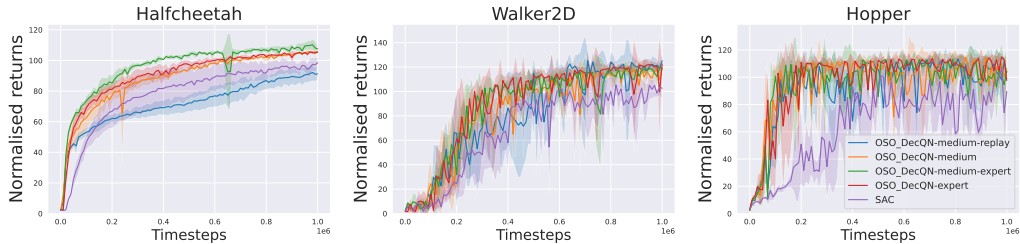

Figure 5: Online learning curves on D4RL, each with corresponding legend. We compare our method of guided learning to the performance of an online agent trained using SAC, without guidance over 1M timesteps. The solid line corresponds to the mean normalised return across 5 seeds with the shaded area corresponding to 1 standard deviation away from the mean.

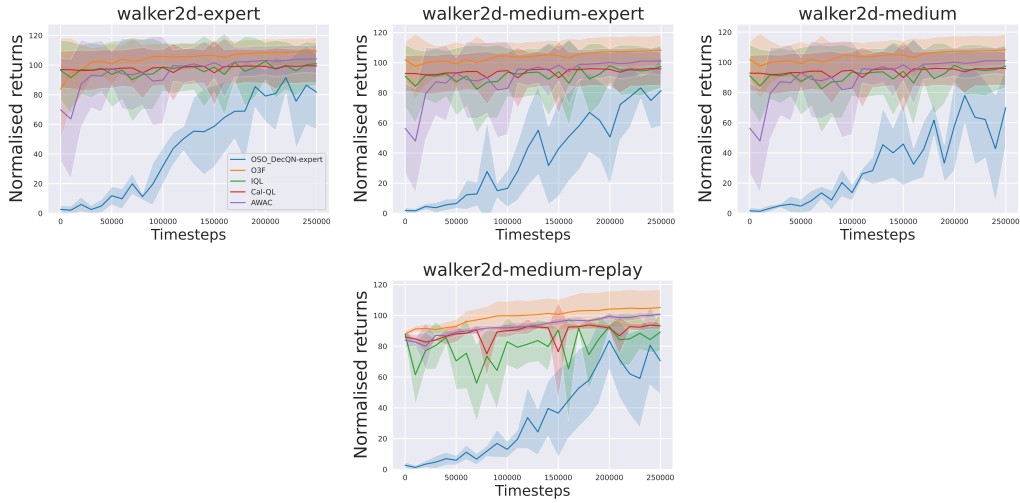

Figure 6: Normalised online returns on Walker2D for different dataset qualities (expert, medium-expert, medium, medium-replay). We compare our action-free method OSO-DecQN, which is pre-trained only on state-only data, to offline-to-online baselines that use full action labels (O3F (Mark et al., 2022), IQL(Kostrikov et al., 2021), CAL-QL (Nakamoto et al., 2023), AWAC (Luo et al., 2023)). Curves show the mean over 5 seeds; shaded regions indicate $\pm$ 1 s.e. over 250k training steps.

## H  COMBINED COMPARISON OF AF-GUIDE AND OSO-DECQN.

Figure 7 overlays the learning curves of AF-Guide, TD3 and OSO-DecQN showing that our action-free method consistently outperforms AF-Guide.

## I  ABLATION EFFECT OF IDM BOOTSTRAPPING TERM.

Table 14 reports an ablation of the IDM bootstrapping term. We compare our full method, which includes the additional loss on offline $(s, s')$ pairs using actions from the current policy, with a variant trained without this term. Result are normalised returns (mean $\pm$ s.e over 5 seeds). The full method generally attains better or comparable final return, yielding a higher total average return (115.8 vs 112.7). This indicates the bootstrapping provides a useful extra signal that improves long-term performance.

## J  LLM USAGE

We use LLMs in our work solely for grammatical purposes.

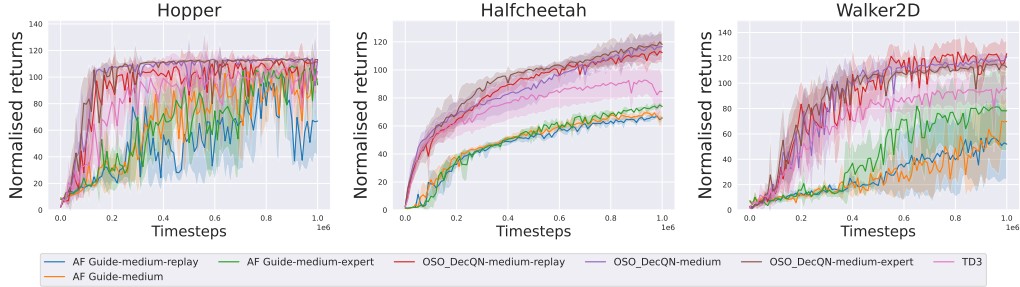

Figure 7: Normalised online returns (mean ± s.e. over 5 seeds) for Hopper, HalfCheetah and Walker2D across different dataset qualities. We plot AF-Guide, TD3 and OSO-DecQN on the same axes. OSO-DecQN consistently achieves comparable or higher final performance than AF-Guide.

Table 14: Effect of IDM relabelling (bootstrapping) on final performance. Normalised returns (mean ± s.e. over 5 seeds) for OSO-DecQN with the full IDM loss and for an ablated variant without the bootstrapping term.

| Dataset | OSO-DecQN (full) | OSO-DecQN (w/o bootstrapping) |
|---|---|---|
| Hopper | | |
| -medium-replay | $111.4 \pm 0.79$ | $113 \pm 0.48$ |
| -medium | $113.2 \pm 1.5$ | $88.1 \pm 9$ |
| -medium-expert | $113.1 \pm 0.46$ | $109.3 \pm 2.5$ |
| -expert | $111.4 \pm 1.7$ | $114.2 \pm 0.64$ |
| Halfcheetah | | |
| -medium-replay | $112.4 \pm 4.1$ | $108.6 \pm 0.61$ |
| -medium | $116.4 \pm 4.7$ | $108.9 \pm 1.6$ |
| -medium-expert | $112.4 \pm 4$ | $119.3 \pm 1.4$ |
| -expert | $113 \pm 4.1$ | $117.6 \pm 2.2$ |
| Walker2d | | |
| -medium-replay | $123.3 \pm 5.4$ | $117 \pm 3.1$ |
| -medium | $120 \pm 5.1$ | $118.1 \pm 2.4$ |
| -medium-expert | $112.6 \pm 1.5$ | $113.2 \pm 2.9$ |
| -expert | $120.7 \pm 2.3$ | $121 \pm 1.4$ |
| Cheetah-Run | | |
| -random-medium-expert | $121.5 \pm 5$ | $124.7 \pm 7$ |
| -medium | $120 \pm 4.4$ | $124.1 \pm 3.5$ |
| -medium-expert | $132.7 \pm 1.43$ | $115.5 \pm 10.2$ |
| -expert | $124.2 \pm 4.9$ | $109.9 \pm 10$ |
| Quadruped-Walk | | |
| -random-medium-expert | $114.2 \pm 5.3$ | $111.8 \pm 6.9$ |
| -medium | $87.4 \pm 13.2$ | $95.6 \pm 13.4$ |
| -medium-expert | $117.5 \pm 2.7$ | $106.9 \pm 7.9$ |
| -expert | $117.7 \pm 1.4$ | $117.2 \pm 2$ |
| Total Average | 115.8 | 112.7 |

