# OpenReview forum: "Action-Free Offline-To-Online RL via Discretised State Policies"
_ICLR.cc/2026/Conference — ICLR 2026 Poster_

### Official Review · Reviewer_EJh5 · 2025-10-28

**Soundness:** 2
**Presentation:** 2
**Contribution:** 2
**Rating:** 6
**Confidence:** 2

**Summary:**

This paper studies a novel and practically motivated setting in reinforcement learning (RL): action-free offline-to-online RL, where the agent must learn from datasets containing only tuples of the form state-reward-next state datasets, without action labels. Such a setting arises naturally in domains like healthcare, finance, and robotics, where action logs may be unavailable due to privacy, storage, or sensor constraints. The paper asks: Can an agent learn useful knowledge from such datasets and transfer it to online learning?
To address this, the authors propose a two-stage framework built around a new offline algorithm, Offline State-Only DecQN (OSO-DecQN), that learns state policies rather than action policies. Instead of predicting actions, the algorithm predicts discretised state differences (i.e., the direction of state change) and uses these predictions to guide online RL.
Empirically, the authors show that OSO-DecQN pre-trained on action-free datasets can accelerate online learning and improve final performance across a range of continuous- and discrete-control tasks (D4RL, DeepMind Control Suite), outperforming existing action-free baselines (e.g., AF-Guide by Zhu et al., 2023). Ablation studies further highlight the importance of discretisation and regularisation, and theoretical results (Theorem 1–2) provide a discretisation-dependent value approximation bound.

**Strengths:**

The paper clearly identifies a real-world gap between existing offline RL methods (which assume full action observability) and practical domains where actions are missing. The authors articulate why action-free learning is challenging and outline an elegant conceptual framework that combines discrete state prediction with conservative regularisation and online guidance. This direction feels both original and valuable for RL research, especially as large unlabelled state datasets become more common.
The proposed OSO-DecQN is conceptually simple yet technically grounded. Overall, the algorithm is a clean extension of DecQN into an action-free paradigm.
The authors provide a series of formal results bounding the approximation error introduced by state discretisation. In particular, Theorem 1 and Theorem 2 derive the difference between the original and discretised optimal value functions, showing that the discretisation granularity controls the value loss. The proofs (Appendix B) are rigorous and grounded in classical contraction and KL-divergence arguments, enhancing the credibility of the approach.

**Weaknesses:**

Although the integration of discretisation, conservative regularisation, and decoupled Q-learning is novel in the action-free context, each component individually builds on well-known ideas (DecQN, CQL-style penalties, inverse dynamics models).
The contribution is therefore more conceptual unification than a fundamentally new RL mechanism.
While the paper claims the IDM is lightweight, its reliability is central to the online guidance mechanism. The IDM must map predicted state differences back to executable actions accurately; errors in this translation could affect results. Although the authors show robustness across architectures (Table 10–11), a more direct comparison of IDM accuracy versus final online performance would better quantify this dependency.
The theoretical analysis focuses entirely on the offline discretisation error.
There is no corresponding guarantee or convergence analysis for the guided online learning phase (e.g., stability of β-switching or IDM-based guidance).
Some formal justification, perhaps connecting it to off-policy improvement or safe exploration, would make the work more complete.

**Questions:**

1.	Since the annealed beta schedule significantly affects results (Fig. 3), analysing it in terms of exploration-exploitation balance could clarify its effect.
2.	It would be better to show a correlation between IDM prediction error and final online performance to support the claim that IDM design is not a major factor.
3.	It would be better if the paper includes comparison of total pretraining + online runtime versus baseline methods to substantiate claims of scalability.

---

> ### Author Response · Authors · 2025-11-18
> **Rebuttal Ejh5**
>
> We thank the reviewer for spending the time to provide a detailed and thoughtful review, and for highlighting both the practical relevance of the action-free offline-to-online setting and the clarity of our theoretical analysis and algorthimic design. We have update the paper (changes marked in red) to incorporate your suggestions, and we hope  we hope that our response effectively address your concerns.
>
> **W**
>
> - **Contribution** We agree that our method builds on several established components (discretisation, conservative regularisation, DecQN, IDM's). Our contribution is not to introduce each ingredient in isolation, but to show that their specific integration in the action-free setting enables the capability of utilising state-only offline data to guide online agents in a more effective manner.
>
> - **Dependence on IDM** We agree that the reliability of the IDM is important for the online guidance mechanism. Based on your suggestion, we have now revised the paper (Appendix E.1), to report the Pearson correlation between IDM prediction error and final online return across the IDM architectures we investigate. The correlations are weak (HalfCheetah-medium: $r=0.19$; Walker2D-medium: $r=-0.09$), indicating that the variation in IDM prediction error explains little of the variance in final performance. Together with Table 10-11 (revised now as 12-13), these results support our claim that, within a reasonable design space, the IDM can be implemented as a lightweight component whose precise architecture is not the main performance driver.
>
> - **Theoretical Analysis** Our theoretical results focus on the offine discretisation error because this is the main source of approximation unique to the action-free setting. The guided online phase itself uses a standard off-policy agent and therefore inherits the same convergence properties as these methods under their usual assumptions. In the revision we provide a more clear qualitative analysis of the role of $\beta$ in terms of the exploration-exploitation balance (Appendix D.3). A full convergence analysis of the combined system is an interesting direction for future work, but we believe the added discussion plus our extensive empirical study across datasets and $\beta$ schedule alleviate potential concerns about stability and justification of the guided phase.
>
> **Q1** **(Beta analysis)**  We thank the reviewer for this suggestion. In the revised paper, we clarify the role of $\beta$ in terms of the exploration-exploitation balance (Appendix D.3). We interpret $\beta$ as controlling how often the agent follows actions suggested by the offline state policy (exploration guided by offline data) versus actions from its own policy (exploitation of the learned value function). A large fixed $\beta=0.8$ encourages strong offline-guided exploration that can prevent the agent from fully exploiting its learned policy, which explains degradation in the performance we observe in HalfCheetah and Walker2D. The annealed schedule starts with a relatively high $\beta$ that gradually allows the agent to increasingly exploit its own policy as learning ensues, yielding both stable learning and better final performance. We have updated Appendix D.3 to make this exploration-exploitation interpretation explicit.
>
> **Q2** **(IDM correlation)** We agree that relating IDM prediction error to online performance can help clarify the role of the IDM. In the revised paper, we now report in Appendix E.1 the Pearson correlation between IDM prediction error and final online return across the IDM architectures we consider. The correlations are weak (HalfCheetah-medium: $r=0.19$; Walker2D-medium: $r=-0.09$), indicating that the variation in IDM prediction error, across different architectures, explains little of the variance in final performance.
>
> **Q3** **(Total compute time)** Thank you for the suggestion. We have added Table 8, which reports the total pretraining plus online runtime of our method and the baseline algorithms TD3/DecQN used for guidance. While our approach incurs an additional offline training cost that is not present when running the baseline, the extra over head during online guidance was small.

---

### Official Review · Reviewer_5mWV · 2025-10-28

**Soundness:** 3
**Presentation:** 2
**Contribution:** 2
**Rating:** 4
**Confidence:** 3

**Summary:**

This paper addresses learning from datasets that lack action labels, containing only (state, reward, next-state) tuples. It proposes OSO-DecQN, which learns policies over discretized states offline and uses them to guide online learning through an inverse dynamics model (IDM).

**Strengths:**

1. The paper tackles practical scenarios where data has no action labels by proposing an offline policy training method that treats a discretized state change as an action and applies offline Q-learning, and demonstrates its effectiveness through various D4RL tasks.

1. The work provides a solution to further improve models pretrained with offline data via online learning by incorporating an IDM (inverse dynamics model), which maps a state change to an action.

**Weaknesses:**

1. Limited comparison with other action-free methods. It seems that in Figure 2, it only compares with the case without pre-training on offline datasets (TD3) and the action-free method (Zhu et al., 2023 [1]). However, given there are more action-free offline training methods [2,3,4], it would be better to compare with all these baselines to fully validate the effectiveness of the proposed action-free training based on state discretization.

2. The presentation of Figure 2 is confusing because although it is for comparing with another action-free offline method (Af-guide), it doesn't directly compare the proposed one and Af-guide. Instead, it is comparing with another baseline (TD3) and indirectly claims that the proposed one is better than AF-guide because it is better than TD3, which I found confusing. It would be better to put all algorithms to be compared in the figure. Also, I wonder if the training/evaluation setup for your method, Af-guide, and TD3 is the same for all experiments.

3. The state discretization seems coarse, which could lose a lot of meaningful state transition information depending on state representation and tasks.

[1] Deyao Zhu, Yuhui Wang, J ̈urgen Schmidhuber, and Mohamed Elhoseiny. Guiding online reinforcement learning with action-free offline pretraining. arXiv preprint arXiv:2301.12876, 2023.

[2] Bohan Zhou, Ke Li, Jiechuan Jiang, and Zongqing Lu. Learning from visual observation via offline pretrained state-to-go transformer. Advances in Neural Information Processing Systems, 36: 59585–59605, 2023.

[3] Hao Luo, Bohan Zhou, and Zongqing Lu. Pre-trained visual dynamics representations for efficient policy learning. In European Conference on Computer Vision, pp. 249–267. Springer, 2024.

[4] Younggyo Seo, Kimin Lee, Stephen L James, and Pieter Abbeel. Reinforcement learning with actionfree pre-training from videos. In International Conference on Machine Learning, pp. 19561–19579. PMLR, 2022.

**Questions:**

1. Zhu et al. 2023 [1] (AF-guide) seems to adopt different approaches for both offline and online training. In Figure 2, is the online and offline training of Af-guide entirely following Zhu et al. 2023? If so, can you provide results for the case of Af-guide (offline) + your method (online) and your method (offline) + Af-guide (online)? This would give a more fine-grained performance comparison of each training process (online/offline).

[1] Deyao Zhu, Yuhui Wang, J ̈urgen Schmidhuber, and Mohamed Elhoseiny. Guiding online reinforcement learning with action-free offline pretraining. arXiv preprint arXiv:2301.12876, 2023.

---

> ### Author Response · Authors · 2025-11-18
>
> We thank the reviewer for the careful and constructive review, and for highlighting both the practical relevance of our setting and the usefulness of combining IDM-based guidance with offline pretraining. Below we address your concerns about comparisons with other action-free methods, the clarity of Figure 2, and the coarseness of the state discretisation as well as your further questions. In response to these points, we have revised the paper to add further plots and clarifications (red text), which we believe improve the overall clarity of our work and results.
>
> **W1**  We thank the reviewer for this suggestion. In the revised paper we have updated our discussion section to explicitly outline the distinction on how our setting differs from the methods you suggested and we summarise the key distinctions here.
>
> Our work focuses on the setting when one has state-only offline data with reward signals, generated by a policy of unknown or mixed quality and the goal is to extract a useful signal to guide an online agent.
>
> As a result, our primary objective is to show that the proposed guidance mechanism consistently improves strong RL baselines. Figure 1 demonstrates that, with guidance, we can improve the performance of TD3 and DecQN across multiple environments and dataset qualities and Figure 5 shows analogous improvements when using SAC. This highlights that our offline-to-online guidance mechanism is not specific to a single RL algorithm.
>
> In contrast, methods [2,3,4] are designed for **video data without rewards, typically assuming expert or near-expert demonstrations**. They rely on imitation-learning objectives to mimic transitions observed in the dataset and do not make use of rewards during training. As such, they target visual imitation from expert videos, which is orthogonal to our focus on pre-training from arbitrary logged data.
>
> Among these works, **[2] can be viewed as a conceptual precursor to AF-Guide [1]**: it trains a decision transformer on a video dataset and uses a similar intrinsic reward mechanism to guide an online agent. AF-Guide adapts this idea to the setting we consider. We therefore include AF-guide as the most relevant instantiation of this family for our domain, which also highlights the increased difficulty of learning when no assumption is made about the quality of the logged data.
>
> [1] Deyao Zhu, Yuhui Wang, Jurgen Schmidhuber, and Mohamed Elhoseiny. Guiding online reinforcement learning with action-free offline pretraining. arXiv preprint arXiv:2301.12876, 2023.
>
> [2] Bohan Zhou, Ke Li, Jiechuan Jiang, and Zongqing Lu. Learning from visual observation via offline pretrained state-to-go transformer. Advances in Neural Information Processing Systems, 36: 59585–59605, 2023.
>
> [3] Hao Luo, Bohan Zhou, and Zongqing Lu. Pre-trained visual dynamics representations for efficient policy learning. In European Conference on Computer Vision, pp. 249–267. Springer, 2024.
>
> [4] Younggyo Seo, Kimin Lee, Stephen L James, and Pieter Abbeel. Reinforcement learning with actionfree pre-training from videos. In International Conference on Machine Learning, pp. 19561–19579. PMLR, 2022.
>
> **W2** **(Concerning Figure 2 plot and experimental setup)** We thank you for your suggestion and have updated our paper to include a figure in Appendix H (Figure 7) showing all three algorithms (our method, AF-guide and TD3) on the same plot for each task, with a clear legend and reference this in the main text (line 366).
>
> Regarding the setup, all methods use the same offline datasets and environment settings. For AF-guide we use the pretrained transformers and guided-learning mechanism provided in the authors' repository, together with their hyperparameters. The only change we make is to increase the number of gradient steps (to 1M) to match our training duration.
>
> **W3**  **(Discretisation method)** We view it as a key contribution of our work that a relatively coarse discretisation can actually be beneficial for learning from action-free data across a wide range of tasks. Furthermore, Theorem 1 provides formal guarantees that discretising the continuous output does not distort the value function arbitrarily and that the approximation error can be made small as the bins are refined.
>
> While there is an inherent bias-variance trade-off in the choice of discretisation, in our experiments (Tables 1 and 2) we find that relatively coarse discretisations already achieve strong performance in the action-free setting and outperform continuous regression methods.
>
> An interesting direction for future work is to adaptively choose the granularity of each state dimension. Doing so would be more challenging than in the action space since state dimensions can have very different ranges and scales, but could further improve the efficiency of our approach.

---

> ### Author Response · Authors · 2025-11-18
> **Rebuttal part 2**
>
> **Q1**  **(Hybrid approach)** Thank you for your suggestion. We confirm that we use the process specified in the original paper to train AF-guide.
>
> We also ran preliminary experiments where we combined our offline state-policy with the AF-guide online procedure. To make this work, we had to adapt their guided online mechanism to use $\Delta s$ predictions produced by our state policy. We observed poor performance on datasets with significant collapse in learning on Half-Cheetah so this hybrid algorithm did not provide a meaningful additional baseline. We therefore only report its final returns, averaged over 5 seeds, in the table below for completeness and contrast it with our method's performance.
>
> | Dataset                 | OSO-DecQN (mean ± se) | OSO-DecQN (offline) + AF-Guide (online) (mean ± se) |
> |-------------------------|------------------------|------------------------------------------------------|
> | Hopper medium-expert    | 113.1 ± 0.46           | 42.5 ± 6.2                                           |
> | HalfCheetah medium-replay | 112.4 ± 4.1          | -8 ± 0.8                                             |
> | Walker2D medium-replay  | 123.3 ± 5.4            | 58 ± 12.5                                            |
> | Walker2d medium         | 121 ± 3.9              | 57.7 ± 16.5                                          |

---

### Official Review · Reviewer_KioB · 2025-10-30

**Soundness:** 2
**Presentation:** 2
**Contribution:** 2
**Rating:** 4
**Confidence:** 3

**Summary:**

This paper proposes a new offline-to-online reinforcement learning (RL) framework designed for action-free offline datasets. The authors introduce a discretized state representation that serves as an action surrogate, enabling the training of a discretized Q-function purely from state transitions. During online learning, an inverse dynamics model (IDM) is trained to map the predicted next-state differences into executable actions. A policy-switching strategy blends the IDM-based actions with the online policy, while a regularization term is used to stabilize and constrain the offline Q-function.
Experiments on D4RL and the Action-Factorized DeepMind Control Suite demonstrate faster convergence and better asymptotic performance for both TD3 and DecQN baselines. Ablation studies further show that both discretization and regularization are crucial for performance.

**Strengths:**

1. The proposed method is conceptually clear and technically sound, addressing a problem where offline datasets lack action labels.
2. The methodology is well-designed, combining existing offline RL techniques with novel discretization and regularization components in a coherent way.
3. The ablation studies are comprehensive and clearly demonstrate the contribution of each component.

**Weaknesses:**

1. The motivation for the action-free offline data setting is not clearly justified. The paper should better explain when and why such data would realistically occur.
2. The examples in the introduction are not entirely convincing. The most plausible application would be robotics from video, but the experiments only consider relatively small state spaces (up to 78 dimensions), which limits the realism of the claim.
3. The baseline coverage is limited — the paper primarily compares against a single online RL algorithm (TD3) and lacks comparison with stronger modern offline-to-online or action-free RL baselines.

**Questions:**

1. Are there plans to extend this framework to visual or multimodal inputs, where the state-space dimension exceeds 1000?
2. Would the method still be effective if the offline dataset and the online environment come from slightly different dynamics distributions?
3. How many offline data samples are used in training, and how does the amount of offline data influence the overall training stability and performance?

---

> ### Author Response · Authors · 2025-11-18
> **Rebuttal (part 1)**
>
> We thank the reviewer for the time spent reviewing our paper and for highlighting the conceptual clarity of our method, coherence of the methodology, and usefulness of the ablation studies. Below we address your concerns regarding the motivation for action-free offline data, generalising to video-based robotics and baseline coverage, and we clarify each of your additional questions in turn. Based on your suggestions we have revised our paper (changes in red) to improve its motivation.
>
> **W1** **(Motivation)** We thank the reviewer for raising this point. Our current goal is to handle datasets where only $(s,r,s')$ tuples are available because action information is missing or cannot be released (e.g. due to noise, privacy, or legal constraints). Based on your suggestion, we have included citations [1, 2, 3, 4] that provide concrete examples of the importance of anonymity of such information in healthcare and finance, which motivate why actions may not be available, to strengthen our claims.
>
> [1] Yu et al., “Reinforcement Learning in Healthcare: A Survey,” ACM Comput. Surv. 2021
>
> [2] Kushida et al., “Strategies for de-identification and anonymization of electronic health record data” J Biomed Inform, 2012
>
> [3] Loukides et al., “Utility-Preserving Transaction Data Anonymization with Low Information Loss,” VLDB J. 2012
>
> [4] Liu et al., “Transactional Data Anonymization for Privacy and Data Mining,” Symmetry 2022
>
> **W2** **(Learning from robotic videos)** We thank the reviewer for bringing this point up and agree that robotics from video is a particularly compelling motivation for action-free learning. Based on your point we have updated the discussion section of our paper to mention this natural next step to our research. We summarise here some of the challenges that need to be overcome to successfully generalise our work to this domain.
>
> **Current work that learn directly from video datasets [5, 6] typically assume expert demonstrations and utilise imitation learning objectives to learn from datasets. In contrast, the offline RL setting we focus in makes no assumptions about the quality of the offline dataset. AF-Guide [7], the benchmark we compare against, can be considered an adaptation of [5] for the offline setting we consider which we benchmark against, demonstrating the challenge of relaxing the assumption of the dataset quality.**
>
> **Furthermore, while many RL algorithms can process high-dimensional inputs, they often struggle as the output dimensionality (typically action dimension) increases.** As a result, most prior work evaluates algorithms in settings where the action dimension typically ranges between 1 and 30. **Our approach learns a state policy, and we demonstrate that it can handle output dimensionalities substantially higher than is standard in the literature.**
>
> A natural next step is to adapt our method to learn from video comprised of suboptimal demonstrations similar to those available in vectorised state format in the D4RL and action-factorised datasets. While we can leverage existing encoders to transform images into vector representations compatible with our approach, additional work is needed on how to reliably extract reward information from visual data to execute RL algorithms which we view as exciting work.
>
> [5] Bohan Zhou, Ke Li, Jiechuan Jiang, and Zongqing Lu. Learning from visual observation via offline pretrained state-to-go transformer. Advances in Neural Information Processing Systems, 36: 59585–59605, 2023.
>
> [6] Younggyo Seo, Kimin Lee, Stephen L James, and Pieter Abbeel. Reinforcement learning with actionfree pre-training from videos. In International Conference on Machine Learning, pp. 19561–19579. PMLR, 2022.
>
> [7]  Deyao Zhu, Yuhui Wang, Jurgen Schmidhuber, and Mohamed Elhoseiny. Guiding online reinforcement learning with action-free offline pretraining. arXiv preprint arXiv:2301.12876, 2023.
>
> **W3** **(Baselines)** We appreciate the concern regarding baseline coverage. The setting we study, offline RL without actions combined with guided online RL, is still relatively new, and there are few directly comparable methods. In the main results of Figure 1, we use **TD3** and **DecQN** as the baseline online algorithms and show that our state policy guidance improves both convergence speed and asymptotic performance across multiple environments and datasets of different qualities. In Figure 5 we also demonstrate analogous gains when using **SAC** as the underlying learner, highlighting that our offline-to-online guidance mechanism is not specific to a single RL algorithm.
>
> Beyond these online RL baselines, we compare against
>
> - (Offline) Imitation learning algorithms (BC on actions, $s'$, $s'-s$, and on $\Delta s$)
> - (Offline) State only DecQN without regularisation
> - **AF-Guide** (Zhu et al., 2023) [7], the only existing action-free guided online method from offline RL that we are aware of, against which our method outperforms on all reported tasks.

---

> ### Author Response · Authors · 2025-11-18
> **Rebuttal (part 2)**
>
> **Q1** **(Extending work)** Please see response to Weakness 2 for details
>
> **Q2** **(different dynamic distribution)** If the dynamics of the online environment differ slightly from those in the offline data, we still expect the method to remain beneficial as long as the offline data continues to provide useful information to guide the agent. Moreover, the policy-switching mechanism allows the base online RL algorithm to override the guidance when it is not helpful, so in the worst case scenario, performance should revert to that of the unguided online learner.
>
> For larger dynamic mismatches, our approach would face the same limitations as other offline RL methods, and additional ideas from domain adaptation would be required.
>
> **Q3** **(Number of offline samples)** We thank the reviewer for pointing this out. In the revised paper, we explicitly report the number of samples in each dataset in Table 3. The dataset sizes range from $2\times10^5$ to $2\times10^6$ transitions (see Table 3 for per environment counts). In general, we find that data quality has a larger impact on performance than the amount of data. For example, the medium dataset contains 1M samples, but because they are generated by a fixed sub-optimal policy, the pre-trained state policy is less effective at guiding the online agent. Across all datasets in Table 3, however, our method remains stable and consistently improves performance over the baselines.

---

### Official Review · Reviewer_cTFm · 2025-11-01

**Soundness:** 3
**Presentation:** 3
**Contribution:** 3
**Rating:** 6
**Confidence:** 3

**Summary:**

In this paper, the authors formalise the setting of action-free offline-to-online RL, where agents must learn from datasets consisting solely of (state, reward, next state) tuples and later leverage this knowledge during online interaction. To address this challenge, they propose learning state policies that recommend desirable next-state transitions rather than actions. First, the authors introduce a simple yet novel state discretisation transformation and propose Offline State-Only DecQN (OSO-DecQN), a value-based algorithm designed to pre-train state policies from action-free data. OSO-DecQN integrates the transformation to scale efficiently to high-dimensional problems while avoiding instability and overfitting associated with continuous state prediction. Second, they propose a novel mechanism for guided online learning that leverages these pre-trained state policies to accelerate the learning of online agents. Together, these components establish a scalable and practical framework for leveraging action-free datasets to accelerate online RL. Empirical results across diverse benchmarks demonstrate that their approach improves convergence speed and asymptotic performance, while analyses reveal that discretisation and regularisation are critical to its effectiveness.

**Strengths:**

1. The motivation for action-free RL is strong, while the explanation for the motivation is clear in the paper.
2. The authors propose an algorithm with novel technical designs.
3. The experimental results are generally good. The ablation study part is helpful.

**Weaknesses:**

1. The paper directly decouples the Q function. Under the case where the real Q function depends on the correlation between different dimensions, could this approximation perform well? Meanwhile, for the argmax of the action, I am wondering what is the choice when $\mathcal{A}$ is not a product space. For instance, if $\mathcal{A}$ is a unit ball, how to define the argmax on each dimension?

2. For the training loss of IDM, could you please provide some motivation for that? Does the success of such approach implicitly depend on the Lipschitzness of the Q value function over actions?

**Questions:**

Please see the weakness

---

> ### Author Response · Authors · 2025-11-18
> **Rebuttal cTFm**
>
> We thank the reviewer for the time spent creating their thoughtful and constructive review, and for highlighting the strengths of the motivation, technical novelty, and experimental study done in our work. Below, we address your concerns and have updated the paper (in red) to clarify the points you raise.
>
> **W1**  We thank the reviewer for raising this point. We have updated the preliminaries to explicitly assume that the action space is a Cartesian product, $\mathcal{A} = \prod_{i=1}^N\mathcal{A}_i$, which allows us to compute the argmax dimension-wise. While the value function factorisation does sacrifice the ability to model strong inter-dimensional correlations in $Q(s,\Delta s)$, it reduces the complexity from exponential in the number of output dimensions to linear, which makes learning feasible on the high-dimensional continuous control tasks we study. Prior work [1] shows that vanilla DQN struggles as the action dimension increases, while DecQN remains stable.
>
> Making such conditional-independence assumptions is very common in ML (e.g. Naive Bayes, mean-field variational inference) and empirically, our results indicate that this trade-off is acceptable, even on tasks that require a substantial amount of coordination between dimensions.
>
> If the action space had additional constraints that meant $\mathcal{A}$ was no longer a product space, an alternative parameterisation (polar coordinates) would be required, or a mapping from the unconstrained argmax to the feasible set.
>
> [1] Seyde, Tim, et al. "Solving continuous control via q-learning." The Eleventh International Conference on Learning Representations. (2023).
>
> **W2**  For the training loss of the IDM, our goal is simply to learn a supervised inverse-dynamics map that is accurate on the states and transitions visited during offline pre-training and guided online learning. Our objective consists of two terms minimised using an L1 regression loss. One term uses data from online transitions, $\|{\bf a} -I_\phi(s,\Delta s)\|_1 $, to update the IDM  and the other term uses data from the offline dataset in the expression
>
> $$\|a_{\text{on,off}}  - I_\phi(s_{\text{off}}, \Delta s_{\text{off}})\|_1$$
>
> where the discrete $\Delta s_{\text{off}}$ is chosen by the pre-trained state policy derived from $Q(s,\Delta s)$. Thus $Q(s,\Delta s)$ is used implicitly in the loss only by selecting which $(s,\Delta s)$ pairs are trained on but the loss itself depends solely on the discrepancy between the IDM predictions and actions. We therefore do not rely on Lipschitzness of the Q-value function over actions. The only regularity we implicitly assume is that the inverse-dynamics mapping ($s,\Delta s) \to a $ is reasonably smooth, which is standard in continuous-control inverse dynamics models.

---

### Official Review · Reviewer_cXZg · 2025-11-04

**Soundness:** 2
**Presentation:** 2
**Contribution:** 3
**Rating:** 4
**Confidence:** 4

**Summary:**

This paper proposes an algorithm for learning value functions from action-free datasets in an offline manner and using these value functions to improve the learning speed in a subsequent online phase. The key parts are discretization method for the state space to assist learning in the offline phase. Experiments with standard offline RL datasets (removing action information) demonstrate the utility of the method.

**Strengths:**

The problem setting is interesting, action-free datasets would seem to be more common in practice and understudied.
The discretization technique for the state space and converting to actions using an IDM is simple and surprisingly effective.

**Weaknesses:**

Generally, there are no glaring issues but some experimental and design choices are unclear to me.
I have included these questions below in the "Questions" section.
Many of these are clarification questions or ablation suggestions and I am happy to increase my score after having more information.

**Questions:**

- Why is DecDQN specifically chosen tobe used rather than a simpler DQN variant given that the method seems to be agnostic to the value-based learning algorithm?


- For the IDM's loss, the $L_1$ loss is used. Why not use the Huber loss if robustness is an issue?

- How does the performance of the method compare to offline-to-online methods that do have access to action information in the offline dataset? We would expect worse performance of course but it would be interesting to see how large the gap really is, it may turn out to be quite small.


- Concerning the discretization step for the state space, it has been found for value learning that regression and predicting a continuous output is inferior to using classification-style losses (the histogram loss) [1]. Have you tried using the histogram loss instead? The loss converts continuous regression targets into a classification target. Then predictions can be done by outputting probabilities over discrete outputs in the support and taking an average. It can be used with a variable number of support points.
This method could also potentially be used for the inverse dynamics model.


- Line 277: Why do we need to use generated actions from the policy to relabel transitions in the offline dataset with the current policy to train the IDM? This would seem to induce a feedback loop since the IDM is trained on its own actions. Could an ablation be run on including this part or not?


- Is the $\epsilon$ hyperparameter in the discretization important? In the appendix, it indicates it is set to be 1e-4 which seems quite small but an ablation showed that this was important to include for some environments.

- How important is the policy-switching strategy? How sensitive is it to the $\beta$ parameter?

- Does the $Q(s, \Delta s)$ value network also get updated during the online phase?


- It is fairly surprising that an inverse dynamics model would be able to use the $\Delta s$ discretized state change effectively since there would be many actions that could map to the same discretized state difference. Why do you think this is effective? Is this exploiting some structure specific to simulated robotics tasks?
An experiment that would help clarify this is to test whether following the action given by the IDM actually leads to the expected dicretized state change. Verifying that the IDM can do this would be interesting.


Minor points (not impacting the score):
- The notation for the discretized state is introduced as $\delta(s,s')$ but in other places it is $\Delta s$ that is used.

- In Theorem 1, M is used to denote the number of bins but it is also used to denote the MDP. It would also be clearer to include the definition of $M$ and $H$ in the theorem statement.

- Table 1: The font is a bit small.

[1] Stop Regression: Training Value Functions via Classification for Scalable Deep RL

---

> ### Author Response · Authors · 2025-11-18
> **Rebuttal (part 1)**
>
> We thank the reviewer for the time spent carefully reading of our paper and for the constructive questions and suggestions. Below, we address each of your points and have updated our paper (in red) and added additional experiments based on your comments that we believe improve both the clarity of our approach and strengthen the motivation.
>
> **Q1**  We chose DecQN because prior work [1] shows that it substantially out performs vanilla DQN as the output (typically actions) dimensionality increases. Standard DQN scales exponentially with the number of discrete output dimensions, whereas DecQN exploits an independence assumption to make this scaling linear and therefore tractable on the more complex environments we consider. While we do take advantage of DecQN's scalability our method is indeed agnostic to the specific discrete Q-learning algorithm.
>
> [1] Seyde, Tim, et al. "Solving continuous control via q-learning." The Eleventh International Conference on Learning Representations. (2023).
>
> **Q2** The Huber loss is indeed a reasonable alternative. In our setting the IDM errors are generally quite small (see Appendix E.2), so training operates almost entirely in the region of the Huber loss where the behaviour is controlled by the threshold $\delta$. This makes performance quite sensitive to the choice of $\delta$. We chose a simpler scale-free $L_1$ loss, which worked well in our experiments, and leave a more systematic study of Huber losses and their thresholds as future work.
>
> **Q3** **(Comparison to action offline to online methods)** When action labels are available, one can first apply standard offline RL methods to train an action policy and then fine-tune this policy directly online using offline-to-online RL methods e.g [2,3,4,5]. We have now added a comparison between our method and these action-based baselines [2,3,4,5] in Appendix G. As expected, methods that can exploit full data achieve higher performance at the very start of online learning than our action-free guided approach.
>
> [2] Mark, Max Sobol, et al. "Fine-tuning offline policies with optimistic action selection." Deep Reinforcement Learning Workshop NeurIPS 2022.
>
> [3] Nakamoto, Mitsuhiko, et al. "Cal-ql: Calibrated offline rl pre-training for efficient online fine-tuning." Advances in Neural Information Processing Systems 36 (2023): 62244-62269.
>
> [4] Nair, Ashvin, et al. "Awac: Accelerating online reinforcement learning with offline datasets." arXiv preprint arXiv:2006.09359 (2020).
>
> [5] Kostrikov, Ilya, Ashvin Nair, and Sergey Levine. "Offline Reinforcement Learning with Implicit Q-Learning." International Conference on Learning Representations.
>
> **Q4** **(Regarding Histogram Loss)** We thank the reviewer for this suggestion and have updated our discussion to include this work. In our method, the discretisation is applied to the output space (state differences) which allows us to exploit the scalable value-based algorithm DecQN, but we still rely on continuous regression for learning the value function. Our method is therefore compatible with this approach and a thorough comparison between regressing the target value and discretising via methods such as HL-Gauss would be an interesting direction for future work.
>
> A practical concern, especially in the offline setting, is that such losses require specifying a support over the Q-values, this typically relies on prior knowledge or sufficiently diverse returns in the dataset, which may be challenging when the data does not span a wide range of returns. We agree that a similar idea could be applied to the inverse dynamics model, and since the action ranges are usually known in advance, discretisation would be more straightforward.

---

> ### Author Response · Authors · 2025-11-18
> **Rebuttal (part 2)**
>
> **Q5** **(IDM loss)** The labelling is used to bootstrap IDM training before we have collected many on-policy transitions in regions covered by the offline dataset. We use the current online policy to propose actions for offline $(s,s')$ pairs so that the IDM can quickly update its predictions to  regions of state space where the offline agent is confident but the online agent has limited samples in its own buffer. We never train the IDM only on its own relabelled actions. These samples are always mixed with the true environment transitions collected online, so the model continues to receive grounded supervision and does not drift. This is conceptually similar to model-based RL methods that roll out trajectories using the current agent's policy under a learned dynamics model for training.
>
> We additionally ran an ablation that removes the bootstrapping term; the early learning curves are similar, but the final asymptotic performance is slightly worse without relabelling. This suggests that relabelling is not critical for initial learning speed, but provides useful additional signal that improves long-term performance.
>
> To make this explicit, we now report a comparison between our full method and the variant without the additional bootstrapping term. The table below shows results for a subset of environments, and the revised paper includes a dedicated appendix section (Appendix I) with results across all datasets. We appreciate the reviewer's suggestion, which strengthens the empirical justification for our design choice.
>
> | dataset quality                 | OSO-DecQN (mean ± se) | OSO-DecQN ablated IDM loss (mean ± se) |
> |---------------------------------|------------------------|------------------------------------------|
> | Quadruped random-medium-expert | 114.2 ± 5.3            | 111.8 ± 6.9                              |
> | Quadruped medium-expert        | 117.5 ± 2.7            | 106.9 ± 7.9                              |
> | Cheetah-run expert             | 124.2 ± 4.9            | 109.9 ± 10                               |
> | Cheetah-run medium-expert      | 132.7 ± 1.43           | 115.5 ± 10.2                             |
> | Hopper medium                  | 113.2 ± 1.5            | 88.1 ± 9                                 |
> | Hopper medium-expert           | 113.1 ± 0.46           | 109.3 ± 2.46                             |
>
> **Q6** **(Importance of epsilon)** Thank you for raising this point, we have revised Appendix D.2 to to clarify this behaviour. The parameter $\epsilon$ is introduced as a tolerance for declaring "no change" in a state dimension after normalisation. This helps avoid spurious sign flips caused by small numerical noise. Our ablation in the appendix shows that performance is quite stable for  a range of small values. Setting $\epsilon=0$ slightly hurts some environments because every tiny fluctuation is treated as a change, whereas making $\epsilon$ very large would coarsen the discretisation too much and can degrade performance.
>
> **Q7** **(Beta analysis)** We provide an experimental analysis of our method to $\beta$ in Appendix D.7. Comparing $\beta$ fixed at various values with varying values to our linearly annealed approach. Our experiments shows that annealing $\beta$ results in more stable and higher asymptotic performance. It also shows that too much exploration (corresponding to high $\beta$) can have a detrimental effect on learning.
>
> **Q8** The pre-trained value network is fixed for the duration of online learning.
>
> **Q9**  **(Regarding mapping discretisations to actions)** We agree that it is not obvious that an inverse dynamics model can reliably map a discretised state change to a useful action. We believe this is feasible in our domains for two reasons. First, each state dimension is discretised into three bins, so in $d_s$ dimensions, we can represent $3^{d_s}$ distinct patterns of state change and since the action dimension is typically much smaller than $d_s$, the many-to-one mapping effect is reduced. Second, the simulated robotics environments we consider have smooth dynamics, so actions that realise the same discretised $\Delta s$ tend to be close in the continuous action space, making it possible to learn mappings through an IDM.
>
> Table 2 reports the error between the $\Delta s$ suggested by our pre-trained state policy and the true $\Delta s$ observed after executing the IDM action in the environment, showing that the realised transitions closely match the targets and that our method is substantially more accurate than the alternative baselines.
>
> ---
>
> **M1** Our intention was to first make explicit that $\Delta s = \delta^\epsilon(s,s') $ denotes a mapping that depends on $\epsilon$. For notational clarity, we then use $\Delta s$ in the remainder of the paper.
>
> **M2** We have updated M for the MDP to use a calligraphic font and updated the paper to include the definitions of M and H
>
> **M3** We have increased the size of the table

---

> > ### Comment · Reviewer_cXZg · 2025-11-26
> > **Reply**
> >
> > Thank you for addressing my questions and clarifying certain aspects.
> > I appreciate the additional experiments and will raise my score.
> >
> > Concerning the histogram loss, I would like to clarify my suggestion as I was unclear. I was not suggesting to change the value estimation algorithm. Instead, I was suggesting an alternative to the state ($\Delta s$) discretization using the histogram representation/loss to do regression.
> >
> > Doing regression on $\Delta s$ does not work well with the squared loss and generally doing regression in this way has been found to be less effective with neural networks [1]. Since the histogram loss involves predicting a probability distribution over a discrete set of points, it seems like a middle ground between standard regression (with squared loss) and fully discretizing (as proposed in the paper under review).
> > I think it may be an interesting alternative to consider.
> >
> > [1] "Investigating the Histogram Loss in Regression"  Imani et al.

---

> > > ### Author Response · Authors · 2025-11-27
> > >
> > > We thank the reviewer again for taking the time to thoroughly review our paper and for the valuable feedback that has helped us improve our work. We agree that using a histogram loss with continuous state targets may provide an appealing middle ground between standard regression and fully discretised state differences, that could mitigate overfitting issues, and we plan to explore this direction in future work. Thank you for the suggestion.

---

### Comment · Area_Chair_itPZ · 2025-11-24
**[ICLR 2026] Author-Reviewer Discussion Phase**

Dear Reviewers,

The authors have posted their rebuttal addressing your concerns. Please kindly review their response, as well as the comments from the other reviewers, and discuss any issues you believe remain unresolved. If the author response does not change your evaluation, please at least provide an acknowledgement indicating that you have carefully reconsidered it.

Thank you again for your dedication and effort in reviewing this submission.
Let’s have a constructive discussion!

Best regards,

Your AC

---

### Author Response · Authors · 2025-12-02
**Review and Discussion Summary (Part 1)**

Despite the circumstances of this reviewing period, we thank the reviewers and AC for the opportunity to engage in discussion that was highly beneficial for our work. To assist the newly assigned AC and help reduce their workload, we summarise below the key points from the reviews and subsequent discussions.

---

**Strengths**

We are grateful that the reviewers acknowledged the importance of the setting we study, and in particular that

- Action-free datasets are common yet understudied, and our method effectively addresses this real-world gap (Reviewer cXZg, 5mWV, Ehj5).
- The motivation for action-free RL is strong (Reviewer cTFm).
- Our approach is conceptually clear, simple, and technically sound, with coherent novel contributions that are technically grounded (Reviewer KioB, cTFm, KioB, Ejh5).
- The direction is original and valuable for RL research as large unlabelled datasets become more common (Reviewer Ejh5).
- The experimental results are strong and the ablation study is helpful and thorough (Reviewer cTFm, KioB).
- Proofs provided are rigorous (Reviewer Ejh5).

---
**Concerns**

During the discussion period, we addressed the reviewers concerns which helped us improve the clarity of our work with additional results and corrections. Specifically

- Clarifcations
  - Reviewer cXZg requested clarifications regarding several experimental and design choices. **We addressed these in our rebuttal, leading them to revise their score from $4\to6$.**
  - Reviewer cTFm requested further clarifications on the choice of the decoupled value function. Empirically, we found **our model's capability to capture correlation is sufficient even on highly coordinated tasks. We also clarified the motivation for the inverse dynamics model (IDM) loss, which maps predicted next states to actions** and does not rely on the Lipschitz continuity of the state-policy Q-values over actions.
  - Reviewer 5mWV raised concerns about the clarity of Figures 1 and 2 when comparing performance. **We addressed this by adding a combined plot** that merges results from both figures.
  - Reviewer Ejh5 requested clarity on several aspects (compute time, IDM correlation, hyperparameters). **We revised our paper to include a compute cost table and additional quantitative metrics** to address their points.

- *Concerns about motivation (KioB)* **We addressed this by revising our paper to include several citations outlining scenarios where actions may not be available in datasets due to privacy or noise-related issues.**

- *Limited comparisons (KioB, 5mWV)* We clarify that prior work in the offline RL action-free setting is limited, with **AF-Guide [1] being the only directly comparable method, that we had already included in the paper.** We also point out that our experiments show **our guidance mechanism can be trained on datasets of various quality and can enhance several online algorithms (TD3, SAC, DecQN)** indicating that our guidance mechanism is not tailored to a single online policy or dataset quality.

   Additionally, **we revised our paper to more clearly distinguish our setting from the related work we cite** that learn from video. We clarify that, **unlike the cited work, we do not assume access to expert datasets**, and explain that **AF-Guide [1] can be viewed as an adaptation of the cited works [2] to our setting**, highlighting the additional challenge relaxing this assumption entails.

- *Coarseness of state discretisation (5mWV)* We outline that **a key contribution of our work is demonstrating that relatively coarse discretisation can be beneficial for learning from action-free datasets**. Additionally, we point out that **our provided formal guarantees indicate that discretising does not distort the value function arbitrarily.**

[1] Deyao Zhu, Yuhui Wang, Jurgen Schmidhuber, and Mohamed Elhoseiny. Guiding online reinforcement learning with action-free offline pretraining. arXiv preprint arXiv:2301.12876, 2023.

[2] Bohan Zhou, Ke Li, Jiechuan Jiang, and Zongqing Lu. Learning from visual observation via offline pretrained state-to-go transformer. Advances in Neural Information Processing Systems, 36: 59585–59605, 2023.

---
**Extra experiments**

*Combining AF-Guide and OSO-DecQN* In response to Reviewer 5mWV's request, we conducted an experiment combining AF-Guide with OSO-DecQN. The preliminary results, which we provided in our rebuttal, showed poor performance, but we believe that the additional design choices required to combine these two different approaches could be further refined in future work.

---

**Recognition from reviewers**

**Only one reviewer (cXZg) got back to us during the discussion phase, confirming that we had addressed their concerns and raised their score from $4\to6$.** Although the remaining reviewers did not reply within the limited discussion window, we believe we have addressed their comments and are grateful for their suggestions, which helped improve our work.

---

> ### Author Response · Authors · 2025-12-02
> **Review and Discussion Summary (Part 2)**
>
> We have, to the best of our ability, summarised the reviewers' comments and our corresponding responses in the hope of easing the AC's workload. We are grateful to the reviewers, AC, SAC, and PC for their time and effort in providing feedback and ensuring that the review process runs smoothly.

---

### Meta-Review · Area_Chair_5hWk · 2026-01-08

**Summary:**

This paper introduces and studies a setting in RL called action-free offline-to-online RL, in which action labels are lacking (e.g., due to privacy or regulatory considerations). For this setting, the paper presents an approach that consists in properly discretizing the state representation (offline phase) and then leveraging an inverse dynamic model (IDM) to map the predicted next-state differences into actions (online phase). The offline part is realized by an algorithm called OSO-DecQN, which indeed the said discretization as an action surrogate for offline training of a relevant Q-function only using state transitions. The paper presents empirical evaluations, demonstrating the effectiveness of the presented approach empirically against TD3 and DecQN baselines on some relevant datasets. The paper includes some preliminary theoretical results (a value approximation error) to support the idea of discretization, but a more thorough treatment is left for future work.

The reviewers unanimously agree the introduced setting is novel and practically relevant in some concrete RL scenarios, and recognized the need for studying such a setting. The presented approach was seen as technically sound and conceptually clean, even though it builds on well-established individual components. The reviewer, while generally positive, raised concerns about ablation studies and possibility of including more relevant baselines. There were questions regarding the effectiveness of the coarse state discretization and that of IDM to output informative actions. Most of these comments were addressed well in the rebuttal. Overall, in view of the reviews and the state of the rebuttal that adequately and sufficiently addresses the key concerns raised by the reviewers, I recommend acceptance.

**Reviewer Concerns:**

__About state discretization.__ Discretizing the state representation constitutes a key underlying idea of the proposed approach. There were concerns regarding the effectiveness of the coarse discretization used in the paper. As the rebuttal clarifies, it was indeed a key observation to empirically validate that a relatively coarse discretization remains highly effective. This is further supported by an approximation error in Theorem 1.

__Presentational issues.__ Some presentational comments were raised by the reviewers, concerning the motivation and requesting clarification of various aspects. These were sufficiently addressed in the rebuttal.


__Comments about empirical evaluation and additional experiments.__ The authors performed additional experiments following the requests or suggestions by the reviewers. One set of experiments was regarding use of a hybrid approach (AF-Guide + OSO-DecQN), which turned out to be inferior. Another set of experiments was done to implement an ablation study, to indicate the effectiveness of the employed bootstrapping term. The results (Appendix I) are highly positive and further strengthens the empirical justification of the design. The reviewers asked questions regarding the choice of included baselines, pointing out some relevant baselines.

__Regarding mapping discretizations to actions.__ A reviewer noted that it is not entirely clear or trivial how the IDM can recover a reliable mapping from the discretized states to informative actions. This point is indeed not fully obvious. As the rebuttal clarifies, the considered robotics domains may not pose a significant issue due to their underlying smooth dynamics. Nevertheless, this suggests that the effectiveness of the IDM relies on assumptions that should be made explicit in the paper. Despite the clarifications in the rebuttal and empirical justifications thereof, I believe this issue is not fully settled, and must be discussed in the paper.

**Reviewer Scores:**

- Reviewer cXZg: The reviewer indicated that the score will be increased (namely from 4 to 6).
- Reviewer cTFm: The reviewer is already positive. Given the state of the rebuttal, I believe the current score would be kept.
- Reviewer KioB: The reviewer's concerns are addressed adequately in the rebuttal. I see it likely that the reviewer would consider increasing the score.
- Reviewer 5mWV: The reviewer's key concerns are about missing baselines and coarse state discretization. The choice of baselines was supported by convincing responses as well as additional experiments. The latter is also addressed in the rebuttal. Although the latter is not fully resolved, in my opinion, without having some (standard) assumptions in place, I believe the reviewer would maintain the score, if not increasing it.
- Reviewer EJh5: The reviewer is already positive. Given the state of the rebuttal, I believe the current score would be kept.

---

### Decision · Program_Chairs · 2026-01-26

Accept (Poster)